# Global Minimizers of Sigmoid Contrastive Loss

**Kiril Bangachev**
kirilb@mit.edu

**Guy Bresler**
guy@mit.edu

**Iliyas Noman**
iliyas@mit.edu

**Yury Polyanskiy**
yp@mit.edu

Department of Electrical Engineering and Computer Science
Massachusetts Institute of Technology
Cambridge, MA, 02139

## Abstract

The meta-task of obtaining and aligning representations through contrastive pre-training is steadily gaining importance since its introduction in CLIP and ALIGN. In this paper we theoretically explain the advantages of synchronizing with *trainable inverse temperature and bias* under the sigmoid loss, as implemented in the recent SigLIP and SigLIP2 models of Google DeepMind. Temperature and bias can drive the loss function to zero for a rich class of configurations that we call $(\mathsf{m}, \mathsf{b}_{\mathsf{rel}})$-Constellations. $(\mathsf{m}, \mathsf{b}_{\mathsf{rel}})$-Constellations are a novel combinatorial object related to spherical codes and are parametrized by a margin $\mathsf{m}$ and relative bias $\mathsf{b}_{\mathsf{rel}}$. We use our characterization of constellations to theoretically justify the success of SigLIP on retrieval, to explain the modality gap present in SigLIP, and to identify the necessary dimension for producing high-quality representations. Finally, we propose a reparameterization of the sigmoid loss with explicit relative bias, which improves training dynamics in experiments with synthetic data. All code is available at RepresentationLearningTheory/SigLIP.

## 1 Introduction

**Background.** *Synchronizing representations* is an increasingly important meta-task in modern machine learning, appearing in several qualitatively different contexts. Models that operate jointly on visual and language data necessitate a synchronization of the representations of images and text [RKH+21, DKAJ21, SRC+21, CSDS21, JYX+21, LLXH22, SBV+22, BPK+22, ZWM+22, HGW+22, WYH+22, CWC+23, ZMKB23b, TGW+25b] and sometimes of additional modalities as well such as audio, thermal data, and others [GENL+23]. State-of-the-art vision models based on self-distillation rely on aligning the representations of augmentations of the same image [CKNH20a, HFW+20, CTM+21]. Likewise, aligning the representations of data produced by teacher and student networks [TKI20b, GS25] has been proposed as a method for distillation. Similarly to the teacher-student setup, the field of backward-compatible learning aims to synchronize the features produced by new models with features of already trained old models [RAVF+22, BPBDB23, SXXS20, JFF+23].

To mathematically formalize the task of synchronizing two representations, suppose that there are $N$ data pairs $\{(X_i, Y_i)\}_{i=1}^N \in (\mathcal{X} \times \mathcal{Y})^{\otimes N}$. For concreteness, one can think of $\mathcal{X}$ as the space of images, $\mathcal{Y}$ as the space of text, and $(X_i, Y_i)$ satisfy a *correspondence relation* such as the fact that they are a true image-caption pair. The goal is to train neural network *encoders* $f_\theta : \mathcal{X} \longrightarrow \mathbb{R}^d$ and $g_\phi : \mathcal{Y} \longrightarrow \mathbb{R}^d$ in such a way that the embeddings produced by them capture the correspondence relation. Such synchronization is usually achieved via minimizing a certain *contrastive loss*.

39th Conference on Neural Information Processing Systems (NeurIPS 2025).

Despite the prevalence of the task of synchronizing representations, there is still limited understanding of *what loss function to use, how to choose its hyper-parameters*, and *what properties of the synchronized embeddings are desirable*. Theoretical results focus mostly on two loss functions – the InfoNCE loss [WI20, CRL$^+$20, RCSJ21, EW22, LS22, PHD20, GRL$^+$24] and the Sigmoid Loss [LCS24, LS22], which both depend on temperature and bias hyper-parameters. While prior works have yielded useful insights, they leave important gaps in our understanding:

*1. Currently understood regimes for the number of represented objects $N$ compared to the dimension of representations $d$ do not reflect practice.* To the best of our knowledge, in all prior theoretical works, either $d \geq N$, or $N$ approaches $+\infty$ for a fixed value of $d$. As a comparison, the SigLIP2 model embeds text and images in $d \approx 10^3$ dimensions and operates with a dataset of size $N \approx 10^{10}$ [TGW$^+$25b]. Thus the practically relevant regime – in which $d \ll N \ll 2^d$ – is not captured by prior work. The different regimes exhibit crucially different behaviors: practically relevant phenomena such as the *modality gap* [LZK$^+$22] only arise when $N > d$, as we show in Theorem 3.6.

*2. The optimal configurations identified by prior works are too rigid.* For example, works in the regime $N \leq d$ typically suggest a simplex structure of the embeddings of each modality [EW22, LS22, LCS24]. This does not explain what the minimizing configurations are *when one modality is pretrained and locked*. In the regime $N \longrightarrow +\infty$, existing results typically suggest a perfect alignment between different representations. Again, this may be too stringent since it has been proposed that "different modalities may contain different information" [HCWI24]. In fact, empirical work suggests that even after synchronization, representations of text and images are completely disjoint, a phenomenon known as *the modality gap* [LZK$^+$22, FMF25].

**Our Contributions.** In the current work, we address these gaps by analyzing the sigmoid loss *with trainable inverse temperature and bias parameters*, as used in Google's SigLIP models [ZMKB23b, TGW$^+$25b] and Gemma 3 [TKF$^+$25]. Making bias and temperature trainable is a key departure from prior theoretical work [WI20, LS22, EW22, LCS24] and leads to novel theoretical guarantees and practical recommendations. We first introduce the sigmoid loss and then describe our contributions.

The sigmoid loss for $U_i = f_\theta(X_i)$ and $V_i = g_\phi(Y_i)$ and inverse temperature[1] $t$ and bias $b$ is:

$$\mathcal{L}^{\mathsf{Sig}}(\theta, \phi; t, b) = \sum_{i=1}^{N} \log\left(1 + \exp(-t\langle U_i, V_i\rangle + b)\right) + \sum_{i \neq j} \log\left(1 + \exp(t\langle U_i, V_j\rangle - b)\right). \quad (1)$$

The first part of the loss encourages the embedding of an image and its caption to be similar, while the second part encourages mismatched image-caption pairs to be dissimilar.

**1. The Geometry of Zero-Loss Configurations.** Our work is the first to rigorously characterize global minima in representation synchronization tasks in the practical regime $N \gg d$.

We show that the SigLIP loss—with trainable temperature and bias—can be driven to zero by a rich family of solutions, which we fully characterize in terms of two novel geometric quantities – the *margin* $\mathsf{m} \geq 0$ and the *relative bias* $\mathsf{b}_{\mathsf{rel}}$. Formally, a $(d, \mathsf{m}, \mathsf{b}_{\mathsf{rel}})$-*Constellation*[2] $\{(U_i, V_i)\}_{i=1}^{N} \in \mathbb{S}^{d-1}$ is defined by the following inequalities:

$$\langle U_i, V_i\rangle \geq \mathsf{m} + \mathsf{b}_{\mathsf{rel}} \qquad \forall i,$$
$$\langle U_i, V_j\rangle \leq -\mathsf{m} + \mathsf{b}_{\mathsf{rel}} \qquad \forall i \neq j. \quad (2)$$

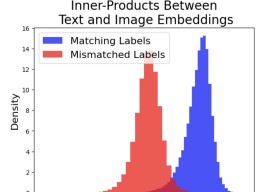

Inner-Products Between Text and Image Embeddings

Figure 1: Distribution of inner products between image and text embeddings from the ImageNet validation set using the $B/16$ $224 \times 224$ SigLIP model available at HuggingFace. The experimental details for all plots as well as further experiments are in Appendix D.

The existence of such $\mathsf{m}, \mathsf{b}_{\mathsf{rel}}$, which one can observe is equivalent to *the inner product separation* $\min_i\langle U_i, V_i\rangle \geq \max_{i\neq j}\langle U_i, V_j\rangle$, is a necessary and sufficient condition for $\{(U_i, V_i)\}_{i=1}^{N}$ to be a global minima of the sigmoid loss with trainable inverse temperature and bias. We show that this is nearly satisfied in practice for the SigLIP model trained on real images and text – See Figure 1. Surprisingly, any configuration satisfying this condition is also a global minimum for the *triplet loss*, see Observation 4. We interpret the margin and relative bias in Section 3.1.

---

[1]Previous works on synchronizing with sigmoid loss such as [ZMKB23b, TGW$^+$25b, LCS24] call $t$ the *temperature*. We call it *inverse temperature* to be more consistent with statistical physics terminology.

[2]We usually omit the parameter $d$ since it is clear from the context and only write "($\mathsf{m}, \mathsf{b}_{\mathsf{rel}}$)-constellation."

In practice, one needs to choose a dimension for the encoders which has large enough "capacity" to hold the embeddings of great many pairs $U_i, V_i$. However, despite intuitive notion that capacity should increase with dimension, to the best of our knowledge no such quantitative characterization was available before our work. Formally, we define the following combinatorial problem and make partial progress in Section 3.2 via a connection to spherical codes.

**Problem 1.** *For a given* $\mathsf{m} \geq 0, \mathsf{b}_{\mathsf{rel}} \in [-1, 1]$, *find the largest number of points* $N = \mathsf{N}_{\mathsf{MRB}}(d, \mathsf{m}, \mathsf{b}_{\mathsf{rel}})$ *such that there exist* $2N$ *vectors* $\{(U_i, V_i)\}_{i=1}^{N} \in \mathbb{S}^{d-1}$ *satisfying* (2).

**2. Success of Zero-Loss Configurations on Downstream Tasks.** In Corollary 1, we use the characterization of zero-loss configurations to show that a standard nearest neighbor search on *any* $(\mathsf{m}, \mathsf{b}_{\mathsf{rel}})$-*Constellation gives perfect retrieval*, even though typically there is no perfect alignment between the two representations. Increasing the margin $\mathsf{m}$ of a constellation makes retrieval robust to larger approximation errors. This is important in practice since retrieval is often performed via an *approximate nearest neighbor search* for computational efficiency [XXL+21, KZ20, MT21].

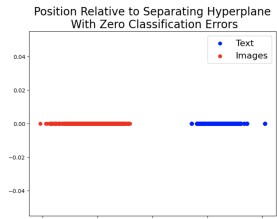

Figure 2: Region of possible $(\mathsf{m}, \mathsf{b}_{\mathsf{rel}})$-Constellations. In red is the impossible region, in which no large configurations are possible (Theorem 3.4). In green is the region where constellations of exponential size exist (Theorem 3.3 and Theorem 3.5). In the shaded region we prove that a modality gap exists (Theorem 3.6).

**3. The Modality Gap: Synchronize, do not Align.** The analysis of [WI20] suggests alignment between representations when training via the InfoNCE loss – the representations of the word "cat" and the image of a cat should (nearly) coincide. Yet, it has been empirically observed that there is a *modality gap* [LZK+22, FMF25]. The representations of images and text – synchronized via the InfoNCE loss in CLIP – do not align, but rather belong to fully disjoint, linearly separable regions. Furthermore, this is not caused by the difference between architecture of image and text encoders, as initially thought, but rather directly by virtue of (approximately) minimizing InfoNCE loss [FMF25].

We shed light on this empirical discovery and prove in Theorem 3.6 that linear separability between modalities holds for any zero-loss configuration of the sigmoid loss in the practically relevant regime $N > d$ when $|\mathsf{b}_{\mathsf{rel}}| < \mathsf{m}$ (Figure 2). We verify our findings by performing experiments with 8 different SigLIP models from Hugging Face on the ImageNet dataset (models given in Table 5). We observe perfect linear separability of image and text embeddings for all models. From a philosophical point of view, as "different modalities may contain different information" [HCWI24], it is only natural that they be represented in disjoint parts of the space.

Figure 3: Modality gap in SigLIP on ImageNet data with the B/16 model with $224 \times 224$ resolution. We find a perfect linear separator using the perceptron algorithm.

We leverage the modality gap to build a linear adapter which can be used towards synchronizing representations when one encoder is locked. This is the reason why we use the name *representation synchronization* rather than representation alignment: alignment between modalities is neither achieved nor desired.

**4. Implications of The Solution Geometry in Practice: Relative Bias Parameterization of Sigmoid Loss.** We propose a parametrization of the sigmoid loss that depends on the *relative bias* rather than the bias in Definition 1. The relative bias parametrization has the following advantages:

*1. Locked Representation:* For example, in LiT [ZWM+22], the image encoder is already trained and locked and we want to synchronize the text encoder with it. The sigmoid loss with trainable parameters in the relative bias parametrization allows us to find a zero-loss configuration for text and images *regardless* of the image encoder. In Observation 1, we show that trainable relative bias and inverse temperature provide a mechanism to implicitly add linear adapters on top of the two encoders as in Figure 4. The adapters we propose in Observations 1 and 2 extend the *Double-Constant Embedding Model* of [LCS24].

*2. More than Two Modalities:* The framework of training with the relative bias parameterization also leads to theoretical guarantees for the global minima of synchronizing more than two modalities via the sigmoid loss. In Observation 2 we show that the parameterization implicitly captures the addition of a modality-dependent linear adapter to each encoder.

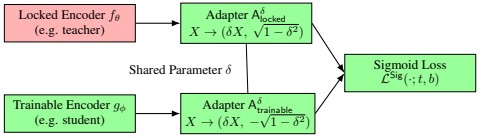

Figure 4: Implicit adapter in relative bias parameterization of sigmoid loss with a locked representation. The parameters $\phi, \delta, t, b$ in green blocks are trainable. Parameter $\theta$ is locked.

*3. Guiding Relative Bias:* Relative bias and margin, which are related by inequalities that we fully characterize in Theorems 3.3 and 3.4, control important properties of the synchronized representations such as *retrieval robustness* and *the presence of a modality gap*. We observe empirically that in the usual sigmoid loss parameterization, Adam [KB15] finds configurations with a zero relative bias, thus limiting the set of trained representations. By adding a relative bias parameter and locking it, we can provably guide the zero loss configuration to a more diverse set of solutions. See Appendix D.4.

## 2 Background and Prior Work

**Representation Learning And Synchronization.** A key insight in [RKH$^+$21, JYX$^+$21] is that training a model to *simultaneously* operate on multiple modalities (such as image and text) enables SOTA performance on individual modalities as well – "if you want to train the best vision model, you should train not just on $N$ images but also on $M$ sentences" [HCWI24]. Several empirical approaches towards synchronizing multiple representations have been proposed, including CLIP [RKH$^+$21], BLIP [LLXH22], ALIGN [JYX$^+$21], LiT [ZWM$^+$22], and SigLIP [ZMKB23b, TGW$^+$25b]. The task of synchronizing representations goes beyond synchronizing across different modalities such as image and text, but also includes synchronizing the representations of a student model to a teacher model with the purpose of distillation [TKI20b, GS25] self-distillation [CTM$^+$21], and aligning the representations of data augmentations [ZIE$^+$16, NF16, GSK18, CKNH20a, HFW$^+$20, CTM$^+$21].

**Formalizing Representation Synchronization.** In this paper, we consider unit-norm encoders $f_\theta : \mathcal{X} \longrightarrow \mathbb{S}^{d-1}, g_\phi : \mathcal{Y} \longrightarrow \mathbb{S}^{d-1}$ (unit norm representations are predominant in practice, see e.g. [SKP15, PVZ15, LWY$^+$17, WXCY17, CKNH20b, HFW$^+$20, TKI20a, ZMKB23b, TGW$^+$25b] and others). Synchronizing the representations produced by $f_\theta, g_\phi$ is usually achieved via minimizing a certain loss function $\mathcal{L}$ over the kernel produced by the embeddings:

$$\mathcal{L}(\theta, \phi; \Gamma) = \mathcal{L}(\{\langle f_\theta(x_i), g_\phi(y_j)\rangle\}_{1 \leq i,j \leq N}; \Gamma) \tag{3}$$

where $\Gamma$ is a set of hyper-parameters, typically involving (inverse) temperature and bias. The optimization is performed via batch first-order optimization methods such as SGD or Adam [KB15].

Depending on the targeted representations, one may choose the parameters over which the optimization is performed. If both $f_\theta, g_\phi$ are untrained, one may perform gradient descent on both $\phi, \theta$ in (3) as in CLIP [RKH$^+$21]. On the other hand, if one of the models – say $f_\theta$ – is already trained and trusted (for example, because it is the teacher model that we are trying to distill [CTM$^+$21, GS25]), we do not update its parameters or only update a small adapter on top of it as in [ZZF$^+$22, LXGY23, GGZ$^+$24, YZWX24, LHY$^+$24, EANP24]. Likewise, this is the case when one of the modalities has already been trained and is locked [ZWM$^+$22, RNP$^+$22, LLXH22, LLSH23].

Besides choosing which parameters to update, one also needs to choose a concrete loss function $\mathcal{L}$. The choices depend on two factors: 1) What is the geometry of the desired minimizing configurations of representations? 2) How efficient is the computation of the loss in terms of the batch size?

We focus on two different loss functions – InfoNCE and sigmoid – and now survey previous works on them. In order to understand the solution geometry of the minimizers, a typical assumption is that the underlying networks $f_\theta, g_\phi$ are sufficiently expressive and can encode any embedding $\{(U_i, V_i)\}_{i=1}^N = \{(f_\theta(X_i), g_\phi(Y_i))\}_{i=1}^N$ [WI20, EW22, LS22, LCS24]. We also adopt this approach.

**Solution Geometry with InfoNCE.** The InfoNCE loss [vdOLV19] with inverse temperature $t > 0$ and bias $b$ is a special case of (3) and takes the following form

$$\mathcal{L}^{\text{InfoNCE}}(\{(U_i, V_i)\}_{i=1}^N; t) = -\frac{1}{N}\sum_{i=1}^N \log \frac{e^{t\langle U_i, V_i\rangle}}{\sum_j e^{t\langle U_i, V_j\rangle}} - \frac{1}{N}\sum_{i=1}^N \log \frac{e^{t\langle U_i, V_i\rangle}}{\sum_j e^{t\langle U_j, V_i\rangle}}. \tag{4}$$

It effectively takes a soft-max over all the rows and columns of the matrix $tU^TV$, an interpretation that yields a connection to the InfoMax principle [HFLM$^+$19] as well as to maximizing point-wise mutual information [TKI20a] and approximate sufficient statistics [OLCM25, LM25].

The solution geometry has been characterized in the case $d \geq N + 1$. The global minimum loss is achieved when $U_i = V_i$ for each $i$ and $U_1, U_2, \ldots, U_n$ form a simplex [EW22, LS22]. When $N \longrightarrow +\infty$, the minimizing measures converge to perfectly aligned ($U_i = V_i$ for all $i$) and uniform (the discrete measure corresponding to $\{U_i\}_{i=1}^n$ converges weakly to the uniform measure). Several works including [SPA$^+$19, EG24] take a different direction and analyze the global minima of the InfoNCE loss (and its symmetrization SimCLR) in terms of performance on downstream (linear) classification tasks instead. While such a geometric characterization is appealing from a practical point of view, these works also do not address the aforementioned gaps in our understanding. The results of [SPA$^+$19] hold in the regime $N \longrightarrow +\infty$ and [EG24] points out that "temperature scaling in the SimCLR loss remains challenging."

In a very different direction, recently it was also rigorously shown that the InfoNCE yields an optimal dimensionality reduction with input data from a Gaussian Mixture Model [BKS25].

**Solution Geometry with Sigmoid Loss.** An alternative loss function used towards alignment is the sigmoid loss [ZMKB23b, TGW$^+$25b] defined in (1). One advantage of the sigmoid loss over InfoNCE is that it does not have a batch normalization term such as $\sum_{j \neq i} \exp(t\langle U_i, V_j\rangle - b)$ and, thus, every pair $(U_i, V_j)$ can be processed separately. This allows for parallel computation.

The solution geometry of configurations achieving global minimum loss has been characterized when $d \geq N$ [LCS24]. For a simplex $\{W_i\}_{i=1}^N$ in $\mathbb{S}^{d-2}$ and some $\delta \in [0, 1]$, it holds that $U_i = (\delta W_i, \sqrt{1 - \delta^2}), V_i = (\delta W_i, -\sqrt{1 - \delta^2})$ for each $i$, where the value of $\delta$ depends on the relationship between $t$ and $b$. In most cases, either the representations collapse to perfectly aligned ($\delta = 1$ and $U_i = V_i$ for all $i$) or antipodal ($\delta = 0$ and $U_i = -V_i = (1, 0, 0\ldots, 0)$ for all $i$). The construction of [LCS24] is the basis for several of our results, including the adapters proposed in Observations 1, 2.

**Other loss functions.** Loss functions such as the triplet loss [SKP15] and $f$-MICl [LZS$^+$23] have also been considered. We further discuss the triplet loss in Appendix A.3.

## 3 Main Results

### 3.1 Geometric Characterization of Zero Loss Representations

In (1), $\mathcal{L}^{\mathsf{Sig}}(\{(U_i, V_i)\}_{i=1}^N; t, b) \geq 0$ holds for any inputs because $\log(1 + e^\kappa) \geq 0$ for any $\kappa \in \mathbb{R}$. Hence, global minimizers are any choice of representations and parameters $\{(U_i, V_i)\}_{i=1}^N; t, b \in (\mathbb{S}^{d-1})^{\otimes N} \times (\mathbb{S}^{d-1})^{\otimes N} \times [0, +\infty] \times [-\infty, \infty]$ leading to a zero loss. We characterize such configurations fully in the following theorems. The proofs are simple and delayed to Appendix A.

**Theorem 3.1** (All Global Minima are $(\mathsf{m}, \mathsf{b}_{\mathsf{rel}})$-Constellations). *Suppose that any iterative algorithm produces a sequence $\{U_i^{(s)}\}_{i=1}^N, \{V_i^{(s)}\}_{i=1}^N, t^{(s)} > 0, b^{(s)}$ for $s = 1, 2, \ldots$ such that*

$$\lim_{s \longrightarrow +\infty} \mathcal{L}^{\mathsf{Sig}}(\{U_i^{(s)}\}_{i=1}^N, \{V_i^{(s)}\}_{i=1}^N; t^{(s)}, b^{(s)}) = 0.$$

*Then, there exists some subsequence indexed by $(s_r)_{r=1}^{+\infty}$ such that*

$$\lim_{r \longrightarrow +\infty} U_i^{(s_r)} = U_i, \qquad \lim_{r \longrightarrow +\infty} V_i^{(s_r)} = V_i \text{ for all } i, \qquad \lim_{r \longrightarrow +\infty} \frac{b^{(s_r)}}{t^{(s_r)}} = \mathsf{b}_{\mathsf{rel}}, \tag{5}$$

*and there exists some $\mathsf{m} \geq 0$ such that $\{(U_i, V_i)\}_{i=1}^N, \mathsf{m}, \mathsf{b}_{\mathsf{rel}}$ satisfy (2).*

**Theorem 3.2** (All $(\mathsf{m}, \mathsf{b}_{\mathsf{rel}})$-Constellations Are Global Minimizers). *Suppose that $\{(U_i, V_i)\}_{i=1}^N \in \mathbb{S}^{d-1}$ satisfies (2) for some $\mathsf{m} > 0$. If we set $b = \mathsf{b}_{\mathsf{rel}} \times t$, then*

$$\lim_{t \longrightarrow +\infty} \mathcal{L}(\{U_i\}_{i=1}^N, \{V_i\}_{i=1}^N; t, \mathsf{b}_{\mathsf{rel}} \times t) = 0.$$

*Moreover, for $\mathsf{m}^* := \frac{1}{2}(\min_i\langle U_i, V_i\rangle - \max_{i \neq j}\langle U_i, V_j\rangle), \mathsf{b}_{\mathsf{rel}}^* := \frac{1}{2}(\min_i\langle U_i, V_i\rangle + \max_{i \neq j}\langle U_i, V_j\rangle),$*

$$\inf_b \mathcal{L}^{\mathsf{Sig}}(\{U_i\}_{i=1}^N, \{V_i\}_{i=1}^N; t, b) = e^{-t\mathsf{m}^* + o(t)}$$

*and is achieved when $b = \mathsf{b}_{\mathsf{rel}}^* \times t + o(t)$.*

This characterization is for global minima in the case of zero loss, which may seem too idealistic. However, it turns out that practically trained models are (up to a small error) also Constellations. In Table 5 in Appendix, we provide the optimal relative bias and margin after removing 5% outliers from the positive and negative pairs.

It turns out that $(\mathsf{m}, \mathsf{b}_{\mathsf{rel}})$-constellations are also global minimizers for the triplet loss, which is another popular contrastive training objective [SKP15], see Appendix A. Theorem 3.2 shows not only that any $(\mathsf{m}, \mathsf{b}_{\mathsf{rel}})$-constellation is a global minimizer, but also that *the optimal margin* $\mathsf{m}^*$ *characterizes the speed of convergence of the loss to zero.*

**Any** $(\mathsf{m}, \mathsf{b}_{\mathsf{rel}})$**-Constellation yields perfect retrieval**    follows as a corollary of Theorem 3.1. In the image-text retrieval task, one is given an image (respectively text) and has to produce the text (respectively image) that best matches it. In our mathematical model, this corresponds to producing $U_i$ on input $V_i$ (and $V_i$ on input $U_i$).

**Corollary 1** (Nearest Neighbor Search Yields Perfect Retrieval). *Suppose that* $\{(U_i, V_i)\}_{i=1}^N$ *is a zero loss configuration. Then, a nearest neighbor of* $U_i$ *among* $\{V_j\}_{j=1}^N$ *is* $V_i$. *If, furthermore, the margin* $\mathsf{m}$ *is strictly positive, this neighbor is unique.*

In practice, retrieval is often performed via *approximate* nearest neighbor search [XXL$^+$21, KZ20, MT21] as this approach has significant computational efficiency advantages. Hence, representations more robust to approximation errors are more desirable. Since $\min_i \langle U_i, V_i \rangle - \max_{i \neq j} \langle U_i, V_j \rangle \geq 2\mathsf{m}$ when (2) is satisfied, representations with a larger margin are more robust. The importance of margin on retrieval has been empirically observed and exploited in several empirical works [LWYY16, DGY$^+$22]. Corollary 1 is for perfect zero-loss constellations. A more robust version also holds, which is closer to practice due to the fact that practical models *are not trained to zero loss* (and cannot be, both due to computational limitations and mislabeled data). We note that in the basic version of the proposition, one can ignore batch size and take $B = N$. We also include the effect of batch size since this is how models are trained in practice.

**Proposition 1** (Robustness of Retrieval via Nearest Neighbor Search). *Let the embedded dataset be* $\{(U_i, V_i)\}_{i=1}^N$. *Suppose that for inverse temperature and bias* $t > 0, b$ *and some* $\xi \in [0, 1]$, *and some batch size* $N > B > \sqrt{N}$, *it holds that:*[3]

$$
\mathbb{E}_{(a_j)_{j=1}^B \sim [N]^{\times k}} \Big[ \sum_{j=1}^B \log \Big( 1 + \exp(-t \langle U_{a_j}, V_{a_j} \rangle + b) \Big)
$$
$$
+ \sum_{i \neq j} \log \Big( 1 + \exp(t \langle U_{a_i}, V_{a_j} \rangle - b) \Big) \Big] \leq \xi \log 2.
$$

*Then, for at least a* $1 - \frac{N\xi}{B(B-1)}$ *fraction of the values* $U_i$ *(respectively,* $V_i$*), a nearest neighbor search returns* $V_i$ *(respectively,* $U_i$*).*

The proof is delayed to Appendix A.2. Note that while $N > B > \sqrt{N}$ is restrictive, it is relevant to models trained with massive compute. For example, in [ZMKB23b], the authors run models with batch sizes up to $64000$ which makes the statement meaningful for datasets of size as large as $10^{10}$.

### 3.2   Constructions of $(d, \mathsf{m}, \mathsf{b}_{\mathsf{rel}})$-Constellations And Cardinality Bounds

Our results so far are vacuous if no $(\mathsf{m}, \mathsf{b}_{\mathsf{rel}})$-Constellations exist. In this section, we show a generic construction which is largely motivated by the Double-Constant Embedding Model of [LCS24] but replaces the simplex with a *spherical code*. For $\alpha \in [-1, 1)$ and $d \in \mathbb{N}$, a $(d, \alpha)$-spherical code is a collection of vectors $X_1, X_2, \ldots, X_N \in \mathbb{S}^{d-1}$ such that $\langle X_i, X_j \rangle \leq \alpha$ for all $i \neq j$ [CSB$^+$13, (52)]. In particular, any $(d, \alpha)$ code is a $(d, \frac{1-\alpha}{2}, \mathsf{b}_{\mathsf{rel}} = \frac{1+\alpha}{2})$-constellation and vice-versa. This implies that any construction of spherical codes immediately implies a construction of $(\mathsf{m}, \mathsf{b}_{\mathsf{rel}})$-Constellations when $\mathsf{m} + \mathsf{b}_{\mathsf{rel}} = 1$. Spherical codes are a well-studied object in combinatorics and many constructions exist depending on $\alpha$ (see [CSB$^+$13] and references therein). The following construction shows that we can extend to the case when $\mathsf{m} + \mathsf{b}_{\mathsf{rel}} \neq 1$.

---

[3]We denote by $[N]^{\times B}$ the uniform distribution over $B$-tuples in $[N]$ of distinct indices.

**Construction 1** (Construction of $(\mathsf{m}, \mathsf{b}_{\mathsf{rel}})$-Constellations)**.** *Consider any* $(d-2, \mathsf{m}, \mathsf{b}_{\mathsf{rel}})$-*constellation* $\{(U_i, V_i)\}_{i=1}^N$. *Then, for any* $\delta, \phi \in [0, 1)$ *such that* $\delta^2 + \phi^2 \leq 1$, *the following vectors form a* $(d, \mathsf{m}', \mathsf{b}'_{\mathsf{rel}})$-*constellation with* $\mathsf{m}' = \delta^2 \mathsf{m}$ *and* $\mathsf{b}'_{\mathsf{rel}} = \delta^2 \mathsf{b}_{\mathsf{rel}} + \phi^2 - (1 - \delta^2 - \phi^2)$:

$$U_i' = (\delta U_i, \phi, \sqrt{1 - \delta^2 - \phi^2}) \qquad and \qquad V_i' = (\delta V_i, \phi, -\sqrt{1 - \delta^2 - \phi^2}). \qquad (6)$$

This construction shows that $(\mathsf{m}, \mathsf{b}_{\mathsf{rel}})$-Constellations not only exist but constitute a rich family. One can in fact construct them from any locked $(d-2)$-dimensional embedding $\{X_i\}_{i=1}^N$ as long as $X_i \neq X_j$ for $i \neq j$ which is the basis of our algorithm for synchronizing with a locked encoder in Observation 1. Recall that the margin impacts the robustness of the representation for retrieval. Thus, it is of both practical and theoretical interest to analyze how large the margin could be for a given dimension $d$ and sample size $N$. Construction 1 immediately gives a recipe for this based on spherical code bounds.

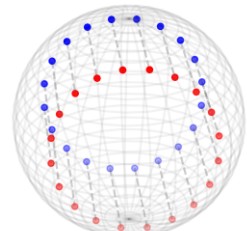

Figure 5: 3D visualization of configurations in Example 3, obtained by minimizing sigmoid loss with Adam; each $(U_i, V_i)$ pair is a reflection across a hyperplane.

That is, let $\mathsf{N}_{\mathsf{SC}}(d, \alpha)$ be the largest possible size of a $(d, \alpha)$-spherical code. Let $\mathsf{E}_{\mathsf{SC}}(\alpha) = \liminf_{d \longrightarrow +\infty} \frac{\log \mathsf{N}_{\mathsf{SC}}(d, \alpha)}{d}$. Determining $\mathsf{N}_{\mathsf{SC}}, \mathsf{E}_{\mathsf{SC}}$ is a well studied problem in coding theory [CSB$^+$13]. We similarly define the numbers $\mathsf{N}_{\mathsf{MRB}}(d, \mathsf{m}, \mathsf{b}_{\mathsf{rel}})$ and $\mathsf{E}_{\mathsf{MRB}}(\mathsf{m}, \mathsf{b}_{\mathsf{rel}})$ for the largest $(\mathsf{m}, \mathsf{b}_{\mathsf{rel}})$-constellations in dimension $d$. Construction 1, together with the classical bound $\mathsf{E}_{\mathsf{SC}}(\alpha) \geq \log_2 \frac{1}{\sqrt{1-\alpha^2}}$ due to Shannon [Sha59] and Wyner [Wyn68] implies:

**Theorem 3.3** (Lower Bound on the Size of Constellations)**.** *Suppose that* $\mathsf{m} \geq 0, \mathsf{b}_{\mathsf{rel}} \in [-1, 1]$ *satisfy* $\mathsf{m} + \mathsf{b}_{\mathsf{rel}} < 1$ *and* $3\mathsf{m} < 1 + \mathsf{b}_{\mathsf{rel}}$. *Then, there exist* $(\mathsf{m}, \mathsf{b}_{\mathsf{rel}})$-*constellations of size exponential in dimension and furthermore*

$$\mathsf{E}_{\mathsf{MRB}}(\mathsf{m}, \mathsf{b}_{\mathsf{rel}}) \geq \mathsf{E}_{\mathsf{SC}}\left(\frac{1 + \mathsf{b}_{\mathsf{rel}} - 3\mathsf{m}}{1 + \mathsf{b}_{\mathsf{rel}} + \mathsf{m}}\right) \geq -\frac{1}{2}\log_2\left(1 - \left(\frac{1 + \mathsf{b}_{\mathsf{rel}} - 3\mathsf{m}}{1 + \mathsf{b}_{\mathsf{rel}} + \mathsf{m}}\right)^2\right).$$

*Proof.* Let $\alpha := \frac{1 + \mathsf{b}_{\mathsf{rel}} - 3\mathsf{m}}{1 + \mathsf{b}_{\mathsf{rel}} + \mathsf{m}} \in [0, 1]$. Let $X_1, X_2, \ldots, X_N$ be an $\alpha$-spherical code in dimension $d - 2$ of size $\exp\left((d-2)(\mathsf{E}_{\mathsf{SC}}(\alpha) + o(1))\right) = \exp\left(d(\mathsf{E}_{\mathsf{SC}}(\alpha) + o(1))\right)$. Choose $\phi, \delta$ as follows: $\delta^2 = \frac{2\mathsf{m}}{1-\alpha}$, $\phi^2 = \frac{2\mathsf{b}_{\mathsf{rel}} + 2 - \delta(3+\alpha)}{4}$. One can easily check that the inequalities $\mathsf{m} + \mathsf{b}_{\mathsf{rel}} \leq 1$ and $3\mathsf{m} \leq 1 + \mathsf{b}_{\mathsf{rel}}$ imply that these values are well-defined in the sense that $\delta^2 > 0, \phi^2 > 0$ and, furthermore, $\delta^2 + \phi^2 \leq 1$. Now, we can apply Construction 1 with $U_i = V_i = X_i$ and $\delta, \phi$ and conclude the desired result. $\square$

The conditions $\mathsf{m} + \mathsf{b}_{\mathsf{rel}} \leq 1$ and $3\mathsf{m} \leq 1 + \mathsf{b}_{\mathsf{rel}}$ are not a virtue of our construction, but turn out to be necessary via an argument resembling Rankin's proof that among any $k + 1$ vectors in $\mathbb{S}^{d-1}$, there exist two with inner product at least $-\frac{1}{k}$ [Ran55]. We actually manage to prove a lower bound for a more general set of configurations than constellations.

**Theorem 3.4** (Upper Bounds on Margin via Relative Bias)**.** *Suppose that* $\{(U_i, V_i)\}_{i=1}^N$ *satisfy that* $\frac{1}{N}\sum_i \langle U_i, V_i \rangle \geq \mathsf{m} + \mathsf{b}_{\mathsf{rel}}$ *and* $\frac{1}{N(N-1)}\sum_{i \neq j} \langle U_i, V_j \rangle \leq -\mathsf{m} + \mathsf{b}_{\mathsf{rel}}$ *(in particular, this holds for any* $(\mathsf{m}, \mathsf{b}_{\mathsf{rel}})$-*constellation). Then, it also holds that*

$$\mathsf{m} + \mathsf{b}_{\mathsf{rel}} \leq 1 \quad and \quad 3\mathsf{m} \leq 1 + \mathsf{b}_{\mathsf{rel}} + o(1).$$

*Proof.* The inequality $\mathsf{m} + \mathsf{b}_{\mathsf{rel}} \leq 1$ is trivial since $\mathsf{m} + \mathsf{b}_{\mathsf{rel}} \leq \max\langle U_1, V_1 \rangle \leq \max \|U_1\|_2 \times \|V_1\|_2 \leq 1$ by (2) and Cauchy-Schwarz. For the second inequality, we use the following fact from [LCS24]. For any unit vectors $\{(U_i, V_i)\}_{i=1}^N$, it holds that $\frac{1}{N^2}\sum_{i \neq j} \langle U_i, V_j \rangle \geq \frac{N-2}{2N^2}\sum_i \langle U_i, V_i \rangle - \frac{1}{2}$. Using $\langle U_i, V_j \rangle \leq \mathsf{b}_{\mathsf{rel}} - \mathsf{m}, \langle U_i, V_i \rangle \geq \mathsf{b}_{\mathsf{rel}} + \mathsf{m}$ gives $(3 - \frac{4}{N})\mathsf{m} \leq 1 + \mathsf{b}_{\mathsf{rel}}$. $\square$

We also provide upper bounds on the size of a constellation given the margin $\mathsf{m}$. This can be used to inform the size of the embedding space given the number of pairs $(U_i, V_i)$ we want to embed in it.

**Theorem 3.5** (Upper Bound on the Size of Constellations)**.** *Suppose that* $\{(U_i, V_i)\}_{i=1}^N$, *is a* $(\mathsf{m}, \mathsf{b}_{\mathsf{rel}})$-*constellation for some* $\mathsf{m} \geq 0, \mathsf{b}_{\mathsf{rel}} \in [-1, 1]$ *which satisfy* $\mathsf{m} + \mathsf{b}_{\mathsf{rel}} \leq 1$ *and* $3\mathsf{m} \leq 1 + \mathsf{b}_{\mathsf{rel}}$. *Then,* $N \leq \exp\left(-d\frac{1}{2}\log\left(1 - \frac{1 + \mathsf{b}_{\mathsf{rel}} - 3\mathsf{m}}{1 + \mathsf{b}_{\mathsf{rel}} + \mathsf{m}}\right) + o(d)\right)$. *Equivalently,* $\mathsf{E}_{\mathsf{MRB}}(\mathsf{m}, \mathsf{b}_{\mathsf{rel}}) \leq -\frac{1}{2}\log\left(1 - \frac{1 + \mathsf{b}_{\mathsf{rel}} - 3\mathsf{m}}{1 + \mathsf{b}_{\mathsf{rel}} + \mathsf{m}}\right)$.

The proof as well as an illustration of the bounds is delayed to Appendix B. In Appendix E, we note that the proof also illustrates a connection with the linear representation hypothesis, e.g. [PCV24].

We note that in [RCSJ21, Theorem 4], the authors find a different connection between spherical codes and minimizers of the InfoNCE loss. In the recent work [WBNL25], the authors show a different dimension lower-bound for existence of vector embeddings for top-$k$ retrieval.

We end with zooming into the Figure 2 in Appendix and plotting the performance of several trained SigLIP models from Hugging Face. We observe two clusters – one composed of the larger so400m models (around 1B parameters) and another of smaller models (up to .4B).

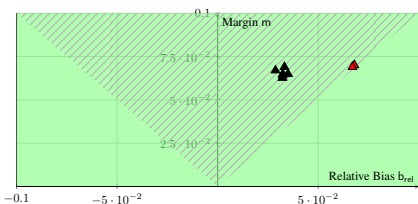

Figure 6: Margin and relative bias with respect to mean inner products of positive pairs $\langle U_i, V_i \rangle$ and negative pairs $\langle U_i, V_j \rangle$ from the ImageNet validation dataset.

## 3.3 The Modality Gap in SigLIP

The construction in Example 1 satisfies the modality gap property – when $\delta > 0$, the representations of the two modalities are separated by a hyperplane (orthogonal to the last coordinate). This phenomenon has been observed empirically on synchronized text and image embeddings in CLIP [LZK+22, FMF25] and in SigLIP by us in Figure 3. We show a rigorous justification for this.

**Theorem 3.6** (Modality Gap in Zero-Loss Configurations). *Suppose that $N \geq d + 2$ and $\{(U_i, V_i)\}_{i=1}^N$ are such that $\langle U_i, V_i \rangle > 0$ for all $i$, $\langle U_i, V_j \rangle < 0$ for all $i \neq j$. This happens for example, when $\mathsf{m} > |\mathsf{b}_{\mathsf{rel}}|$ in a $(\mathsf{m}, \mathsf{b}_{\mathsf{rel}})$-Constellation. Then, there exists some $h \in \mathbb{S}^{d-1}$ such that:*

$$\langle h, U_i \rangle > 0 \qquad \text{for all } i, \tag{7}$$

$$\langle h, V_j \rangle < 0 \qquad \text{for at least } N - d \text{ values of } j. \tag{8}$$

The fact that the condition (8) is satisfied for at least $N - d$ values of $j$ instead of all $N$ is rather minor in practice. As mentioned, in SigLIP2 [TGW+25b], $N \approx 10^{10}$ and $d \approx 10^3$. Thus, our result shows the modality gap holds for all but .0000001% of the text embeddings. We note that the theorem is essentially tight – in Example 3, we show an example in which $d - 1$ of the vectors $V_j$ cannot be separated from the vectors $U_i$. We also note that $\mathsf{m} > |\mathsf{b}_{\mathsf{rel}}|$ is also plausible as practically trained models such as SigLIP2 have a small relative bias of magnitude less than 0.1 [ZMKB23a].

*Proof Sketch.* The full proof is in Appendix C, where we also analyze further properties of configurations satisfying (7) and (8). Here we give a sketch. First, we use Helly's theorem (Theorem C.1) to show that the convex sets $\{x : \langle x, U_i \rangle > 0\}_{i=1}^N$ have a non-empty intersection and, hence, there exists some $h \in \mathbb{S}^{d-1}$ such that $\langle h, U_i \rangle > 0$ for each $i$. Then, we use the hyperplane separation theorem (Theorem C.3) to show that the projection $\bar{h}$ of $h$ on the convex cone defined by $U_1, U_2, \ldots, U_N$ also has this property. Finally, we use Caratheodory's theorem (Theorem C.2) to show that $\bar{h}$ has a positive inner product with all the vectors $U_i$ and is in the convex cone of at most $d$ of the vectors $U_j$. This implies that $\bar{h}$ has a negative inner product with all the other $N - d$ vectors $V_k$. $\qquad\square$

## 3.4 Experiments: Sigmoid Loss with Explicit Relative Bias Parameterization

Due to the importance of the relative bias parameters for global minima of sigmoid loss, we propose a parameterization that explicitly captures this dependence.

**Definition 1** (Parameterization with Explicit Relative Bias). *The relative bias parametrization of the sigmoid loss for encoder $f_\theta, g_\phi$ over data pairs $\{(X_i, Y_i)\}_{i=1}^N$ with $U_i = f_\theta(X_i), V_i = g_\phi(Y_i)$ is*

$$\mathcal{L}^{\mathsf{RB-Sig}}(\theta, \phi; t, \mathsf{b}_{\mathsf{rel}}) = \sum_{i=1}^N \log\left(1 + e^{-t\langle U_i, V_i \rangle + t\mathsf{b}_{\mathsf{rel}}}\right) + \sum_{i \neq j} \log\left(1 + e^{t\langle U_i, V_j \rangle - t\mathsf{b}_{\mathsf{rel}}}\right).$$

Clearly, $\mathcal{L}^{\mathsf{RB-Sig}}(\theta, \phi; t, \mathsf{b}_{\mathsf{rel}}) = \mathcal{L}^{\mathsf{Sig}}(\theta, \phi; t, \mathsf{b}_{\mathsf{rel}} \times t)$ so the loss functions are the same. However, we show that running Adam [KB15] on $\mathcal{L}^{\mathsf{RB-Sig}}$ yields faster convergence – see Figures 7 and 8. It also provides the additional flexibility to freeze the relative bias to a desired value and only train inverse temperature. This may be important since we observe that in practice relative bias converges

to 0 when not frozen. For example, in SigLIP2 [ZMKB23a] with the B/16 model with resolution $384 \times 384$, we have learned parameters $t \approx 117.8, b \approx -12.9$, so $\mathsf{b_{rel}} \approx -0.11$. We give further experimental evidence for this in Appendix D.4. Thus, we propose using our parameterization $\mathcal{L}^{\mathsf{RB-Sig}}$ in practice over $\mathcal{L}^{\mathsf{Sig}}$.

**Fixed Relative Bias.**    One functionality that this new parameterization gives us is to train models with a fixed relative bias. As expected from Fig. 2, this in turn has an effect on the margin of the configurations. We plot in Fig. 10 in Appendix the evolution of margins (computed as $(\min_i \langle U_i, V_i \rangle - \max_{i \neq j} \langle U_i, V_j \rangle)/2$) for different fixed relative biases and in Fig. 9 the final optimal relative biases (computed as $(\min_i \langle U_i, V_i \rangle + \max_{i \neq j} \langle U_i, V_j \rangle)/2$).

**Locked Encoder.**    In particular, this gives a concrete recipe for training via the sigmoid loss with a fixed representation: one just adds a simple adapter $\mathsf{A}^\delta_{\mathsf{locked}}$ that transforms $X_i \longrightarrow (\delta X_i, \sqrt{1-\delta^2})$ for the locked representation and $\mathsf{A}^\delta_{\mathsf{trainable}}$ that transforms $X_i \longrightarrow (\delta X_i, -\sqrt{1-\delta^2})$ for the trainable representation. It turns out that the relative bias parametrization captures this transformation *without explicitly adding an adapter*.

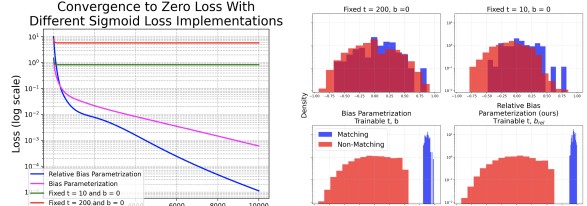

Figure 7: Synchronizing a locked representation with different sigmoid loss functions on synthetic data. On the left, we have the evolution of the loss function. On the right, we show the distributions of non-matching inner products $\langle U_i, V_j \rangle$ for $i \neq j$ in red and matching inner products $\langle U_i, V_i \rangle$ in blue for each model.

**Observation 1.** *For any* $\{(U_i, V_i)\}_{i=1}^N$ *and* $\delta, \mathsf{b_{rel}}, t$, *it is the case that*

$$\mathcal{L}^{\mathsf{RB-Sig}}(\{(\mathsf{A}^\delta_{\mathsf{lock}}(U_i), \mathsf{A}^\delta_{\mathsf{train}}(V_i))\}_{i=1}^N; t, \mathsf{b_{rel}}) = \mathcal{L}^{\mathsf{RB-Sig}}(\{(U_i, V_i)\}_{i=1}^N; t\delta^2, \frac{\mathsf{b_{rel}} + (1-\delta)^2}{\delta^2}).$$

As we can see in Figure 7, the models with trainable $t, b$ (respectively $t, \mathsf{b_{rel}}$) significantly outperform the model with fixed temperature and bias. Furthermore, the convergence to zero loss is faster for $\mathcal{L}^{\mathsf{RB-Sig}}$ than $\mathcal{L}^{\mathsf{Sig}}$. Thus, we recommend synchronizing with $\mathcal{L}^{\mathsf{RB-Sig}}$ and *trainable* $t, \mathsf{b_{rel}}$.

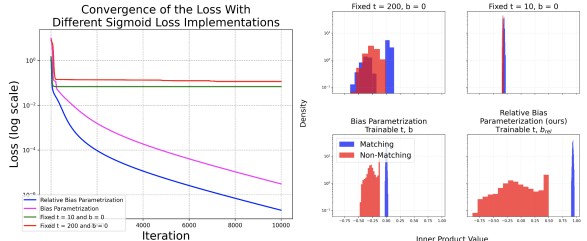

Figure 8: Inner product distributions between $k = 4$ modalities synchronized with different implementations of sigmoid loss. We plot the same data as in 7.

**More Modalities and A New Perspective on Simplex Embeddings.**    Our discussion so far has been predominantly in the case of two modalities. To synchronize the representations $\{(U_i^{(1)}, \ldots, U_i^{(k)})\}_{i=1}^N$ of $k > 2$ modalities, one typically minimizes the sum of several pairwise losses [TKI20a, GENL$^+$23]. More formally, if $G = (V, E)$ is the *synchronization graph* on vertex set $V = \{1, 2, \ldots, k\}$ the different modalities, one minimizes

$$\sum_{(j_1, j_2) \in E} \mathcal{L}^{\mathsf{RB-Sig}}(\{(U_i^{(j_1)}, U_i^{(j_2)})\}_{i=1}^N; t, \mathsf{b_{rel}}). \tag{9}$$

Common instances are when $G$ is the complete graph and one sums over all pairwise losses [TKI20a], and when $G$ is a star graph with one central modality [GENL$^+$23]. Since the loss function $\mathcal{L}^{\mathsf{Sig}}$ is non-negative, a configuration $\{(U_i^{(1)}, \ldots, U_i^{(k)})\}_{i=1}^N$ is *zero-loss* if and only if there exist some $\mathsf{m}, \mathsf{b_{rel}}$ such that $\{(U_i^{(j_1)}, U_i^{(j_2)})\}_{i=1}^N, \mathsf{m}, \mathsf{b_{rel}}$ is zero loss for any $(j_1, j_2) \in E$. In particular, $\{(U_i^{(1)}, \ldots, U_i^{(k)})\}_{i=1}^N$ is zero loss if there exist some $\mathsf{m}, \mathsf{b_{rel}}$ such that $\{(U_i^{(j_1)}, U_i^{(j_2)})\}_{i=1}^N, \mathsf{m}, \mathsf{b_{rel}}$ is zero loss for all $j_1 \neq j_2$. This leads us to the following construction.

**Construction 2** (Construction of Constellations). *Consider any* $(d - k + 1, \alpha)$*-code* $\{X_i\}_{i=1}^N$. *Let* $w_1, w_2, \ldots, w_k \in \mathbb{S}^{k-1}$ *be the vertices of a regular $k$-simplex. Then, for any $\delta \in [0, 1)$, the following configuration is zero loss for any synchronization graph:*

$$U_i^{(j)} = (\delta X_i, \sqrt{1-\delta^2} w_j), \qquad \mathsf{m} = \frac{\delta^2(1-\alpha)}{2}, \qquad \mathsf{b_{rel}} = \frac{\delta^2(1-\alpha)}{2} - \frac{1-\delta^2}{k-1}. \tag{10}$$

We can enforce this structure with an adapter which appends a modality-dependent suffix.

**Observation 2.** *For any* $\{(U_i^{(1)}, U_i^{(2)}, \ldots, U_i^{(k)})\}_{i=1}^N$ *and* $\delta, \mathsf{b_{rel}}, t,$

$$\mathcal{L}^{\mathsf{RB-Sig}}\Big(\{\mathsf{A}_1^\delta(U_i^{(1)}), \ldots, \mathsf{A}_k^\delta(U_i^{(k)})\}_{i=1}^N; t, \mathsf{b_{rel}}\Big) = \mathcal{L}^{\mathsf{RB-Sig}}\Big(\{U_i^{(1)}, \ldots, U_i^{(k)}\}_{i=1}^N; t\delta^2, \mathsf{b_{rel}} + \frac{(1-\delta)^2}{k-1}\Big).$$

### 3.5 Ablation Studies

*Training the temperature and bias* is the key mechanism that drives the loss to zero for a wide range of configurations which we described as $(\mathsf{m}, \mathsf{b_{rel}})$-Constellations. We compared our proposal of training $\mathcal{L}^{\mathsf{RB-Sig}}$ with trainable $\mathsf{b_{rel}}, t$ parameters against several alternatives. We concretely focus on the *inner product separation condition* $\min_i \langle U_i, V_i \rangle \geq \max_{i \neq j} \langle U_i, V_j \rangle$ and corresponding margin. This is a key property of interest since, as explained, it determines the success of the model on retrieval via (approximate) nearest neighbor search. We also analyze the convergence of the loss to zero, which is an indicator for how many epochs of training a model needs till convergence.

**1. Training with fixed low inverse temperature ($t \lesssim 10$) and bias** as in the analysis of [LCS24] is a first natural alternative. We observed that in the contexts of synchronizing multiple embeddings (Figure 8) and synchronizing with a locked encoder (Figure 7), the resulting embeddings fail to satisfy the inner product separation condition or do so with a much smaller margin than models with trainable inverse temperature and (relative) bias.

**2. Training with fixed high inverse temperature ($t \gg 10$) and bias.** Any $(\mathsf{m}, \mathsf{b_{rel}})$-constellation is nearly a global minimum in the regime of large $t$, so one may expect similar performance to the trainable inverse temperature and bias model. This approach fails in practice since it does not allow the algorithm to gradually find the synchronized representations. The embeddings discovered are not useful towards retrieval as the inner-product separation fails and the loss does not approach zero (Figures 7 and 8), even though representations with nearly-zero loss exist due to the low temperature.

**3. Training with bias parameterization.** Finally, we compared against the bias parameterization $\mathcal{L}^{\mathsf{Sig}}$ used in [ZMKB23b, TGW$^+$25b]. While this model also generally led to $(\mathsf{m}, \mathsf{b_{rel}})$-Constellations which can be used for perfect retrieval, we observed *slower convergence of the loss function* and *smaller margin* due to a tendency towards zero relative bias (Figures 7 and 8).

## 4 Limitations and Future Directions

We provide the first theoretical analysis of synchronizing representations in the practically relevant regime $d \ll N \ll 2^d$. A theoretical limitation is that while we identify global minimizers and empirically show that first-order methods such as Adam find them, we do not prove rigorous performance guarantees for first-order methods. Another theoretical limitation is that we do not fully resolve the combinatorial Problem 1, which as we point out is practically relevant for choosing the embedding dimension of encoders. Finally, we show that the parametrization of sigmoid loss with relative bias leads to more flexibility and faster convergence on synthetic data, but do not perform experiments with it on real data. We believe that all of these are exciting directions for future research.

## Acknowledgments

KB was supported in part by an NAE Grand Challenge Vest Fellowship. GB was supported by NSF award 2428619. The research of YP was supported, in part, by the United States Air Force Research Laboratory and the United States Air Force Artificial Intelligence Accelerator under Cooperative Agreement Number FA8750-19-2-1000. The views and conclusions contained in this document are those of the authors and should not be interpreted as representing the official policies, either expressed or implied, of the United States Air Force or the U.S. Government. The U.S. Government is authorized to reproduce and distribute reprints for Government purposes notwithstanding any copyright notation herein.

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

# A  Omitted Proofs From Section 3.1

## A.1   Global Minimizers of The Sigmoid Loss are $(\mathsf{m}, \mathsf{b}_{\mathsf{rel}})$-Constellations

We will repeatedly use the following fact which follows from $\log(1 + \exp(\kappa)) \geq 0$ for any $\kappa \in \mathbb{R}$.

**Observation 3.** *For any $\{(U_i, V_i)\}_{i=1}^N$ and $t, b$, it holds that*

$$\max\left(\max_i \log\left(1 + \exp(-t\langle U_i, V_i\rangle + b)\right), \max_{i \neq j} \log\left(1 + \exp(t\langle U_i, V_j\rangle - b)\right)\right) \tag{11}$$

$$\leq \mathcal{L}^{\mathsf{Sig}}(\{U_i\}_{i=1}^N, \{V_i\}_{i=1}^N; t, b)$$

$$\leq N^2 \times \max\left(\max_i \log\left(1 + \exp(-t\langle U_i, V_i\rangle + b)\right), \max_{i \neq j} \log\left(1 + \exp(t\langle U_i, V_j\rangle - b)\right)\right). \tag{12}$$

*Proof of Theorem 3.1.*  Suppose that $\lim_{s \longrightarrow +\infty} \mathcal{L}^{\mathsf{Sig}}(\{U_i^{(s)}\}_{i=1}^N, \{V_i^{(s)}\}_{i=1}^N; t^{(s)}, b^{(s)}) = 0$ indeed holds. By, (11), this means that

$$\lim_{s \longrightarrow +\infty} \log\left(1 + \exp(-t^{(s)}\langle U_i^{(s)}, V_i^{(s)}\rangle + b^{(s)})\right) = 0 \qquad \forall i,$$

$$\lim_{s \longrightarrow +\infty} \log\left(1 + \exp(t^{(s)}\langle U_i^{(s)}, V_j^{(s)}\rangle - b^{(s)})\right) = 0 \qquad \forall i \neq j.$$

Equivalently,

$$\lim_{s \longrightarrow +\infty} -t^{(s)}\langle U_i^{(s)}, V_i^{(s)}\rangle + b^{(s)} = -\infty \qquad \forall i,$$

$$\lim_{s \longrightarrow +\infty} t^{(s)}\langle U_i^{(s)}, V_j^{(s)}\rangle - b^{(s)} = -\infty \qquad \forall i \neq j.$$

Equivalently,

$$\lim_{s \longrightarrow +\infty} t^{(s)}\left(\langle U_i^{(s)}, V_i^{(s)}\rangle - \frac{b^{(s)}}{t^{(s)}}\right) = +\infty \qquad \forall i,$$

$$\lim_{s \longrightarrow +\infty} t^{(s)}\left(\langle U_i^{(s)}, V_j^{(s)}\rangle - \frac{b^{(s)}}{t^{(s)}}\right) = -\infty \qquad \forall i \neq j.$$

In particular, as $t^{(s)} > 0$ always, this means that for all large enough $s$, the quantity $\mathsf{b}_{\mathsf{rel}}^{(s)} := \frac{b^{(s)}}{t^{(s)}}$ satisfies that

$$\langle U_i^{(s)}, V_i^{(s)}\rangle - \mathsf{b}_{\mathsf{rel}}^{(s)} \geq 0 \qquad \forall i, \tag{13}$$

$$\langle U_i^{(s)}, V_j^{(s)}\rangle - \mathsf{b}_{\mathsf{rel}}^{(s)} \leq 0 \qquad \forall i \neq j. \tag{14}$$

However, as all $U_i^{(s)}, V_i^{(s)}$ are unit vectors, $\langle U_i^{(s)}, V_i^{(s)}\rangle \leq 1$, $\langle U_i^{(s)}, V_j^{(s)}\rangle \geq -1$ holds for any $i, j, s$. Hence, for all large enough $s$, (13) and (14) imply that

$$-1 \leq \mathsf{b}_{\mathsf{rel}}^{(s)} \leq 1.$$

Now, observe that $\{U_1^{(s)}, U_2^{(s)}, \ldots, U_N^{(s)}, V_1^{(s)}, V_2^{(s)}, \ldots, V_N^{(s)}, \mathsf{b}_{\mathsf{rel}}^{(s)}\} \in (\mathbb{S}^{d-1})^{\otimes 2N} \times [-1, 1]$. As $(\mathbb{S}^{d-1})^{\otimes 2N} \times [-1, 1]$ is a compact set, $\{U_1^{(s)}, U_2^{(s)}, \ldots, U_N^{(s)}, V_1^{(s)}, V_2^{(s)}, \ldots, V_N^{(s)}, \mathsf{b}_{\mathsf{rel}}^{(s)}\}_{s=1}^{+\infty}$ has a convergent subsequence. Suppose that it converges to $\{U_1, U_2, \ldots, U_N, V_1, V_2, \ldots, V_N, \mathsf{b}_{\mathsf{rel}}\}$. By (13) and (14), we have that

$$\langle U_i, V_i\rangle - \mathsf{b}_{\mathsf{rel}} \geq 0 \qquad \forall i, \tag{15}$$

$$\langle U_i, V_j\rangle - \mathsf{b}_{\mathsf{rel}} \leq 0 \qquad \forall i \neq j. \tag{16}$$

Setting

$$\mathsf{m} = \min\left(\min_i \langle U_i, V_i\rangle - \mathsf{b}_{\mathsf{rel}}, \min_{i \neq j} \mathsf{b}_{\mathsf{rel}} - \langle U_i, V_j\rangle\right)$$

gives the desired result. □

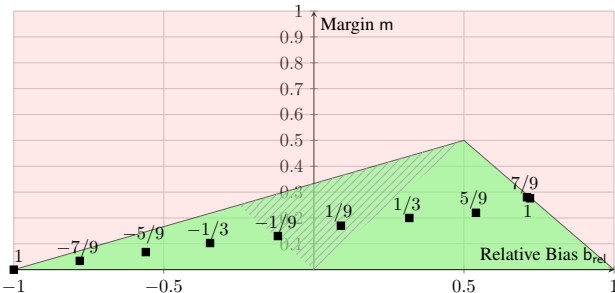

Figure 9: Achieved optimal relative bias and margin when training with a fixed relative bias. The annotations correspond to the fixed relative bias.

*Proof of Theorem 3.2.* Suppose that $\{(U_i, V_i)\}_{i=1}^N$ are a $(\mathsf{m}, \mathsf{b}_{\mathsf{rel}})$-Constellation with some $\mathsf{m} > 0$. Then, for any $t$, it holds that

$$t(\langle U_i, V_i \rangle - \mathsf{b}_{\mathsf{rel}}) \geq \mathsf{m} \times t \qquad \forall i, \tag{17}$$

$$t(\langle U_i, V_j \rangle - \mathsf{b}_{\mathsf{rel}}) \leq -\mathsf{m} \times t \qquad \forall i \neq j. \tag{18}$$

In particular, by Observation (12), it follows that

$$\mathcal{L}^{\mathsf{Sig}}(\{U_i\}_{i=1}^N, \{V_i\}_{i=1}^N; t, t\mathsf{b}_{\mathsf{rel}})$$

$$\leq N^2 \times \log\left(1 + \exp(-\mathsf{m} \times t)\right)$$

$$\leq N^2 \exp(-\mathsf{m} \times t) = \exp(-\mathsf{m} \times t + 2\log N/t) = \exp(-\mathsf{m} \times t + o(t))$$

which proves the convergence to zero. Choosing $(\mathsf{m}^*, \mathsf{b}_{\mathsf{rel}}^*)$ in this argument, where

$$\mathsf{m}^* := \frac{1}{2}(\min_i \langle U_i, V_i \rangle - \max_{i \neq j} \langle U_i, V_j \rangle) > \frac{1}{2}((\mathsf{m} + \mathsf{b}_{\mathsf{rel}}) - (-\mathsf{m} + \mathsf{b}_{\mathsf{rel}})) = \mathsf{m} > 0,$$

$$\mathsf{b}_{\mathsf{rel}}^* := \frac{1}{2}(\min_i \langle U_i, V_i \rangle + \max_{i \neq j} \langle U_i, V_j \rangle),$$

also proves that $\inf_b \mathcal{L}^{\mathsf{Sig}}(\{U_i\}_{i=1}^N, \{V_i\}_{i=1}^N; t, b) \leq e^{-t\mathsf{m}^* + o(t)}$.

All that is left to show is that

$$\inf_b \mathcal{L}^{\mathsf{Sig}}(\{U_i\}_{i=1}^N, \{V_i\}_{i=1}^N; t, b) \geq e^{-t\mathsf{m}^* + o(t)}.$$

Equivalently, we can show that $\inf_b \mathcal{L}^{\mathsf{Sig}}(\{U_i\}_{i=1}^N, \{V_i\}_{i=1}^N; t, b) \geq \log\left(1 + e^{-t\mathsf{m}^* + o(t)}\right)$ since $\log(1 + \gamma) = \gamma + o(\gamma)$ as $\gamma \longrightarrow 0$. However, by (11), for the last inequality, it is enough to show that

$$\max\left(\max_i\left(-t\langle U_i, V_i \rangle + b\right), \max_{i \neq j}\left(t\langle U_i, V_j \rangle - b\right)\right) \geq -t \times \mathsf{m}^*.$$

Equivalently

$$\min\left(\min_i\left(\langle U_i, V_i \rangle - \frac{b}{t}\right), \min_{i \neq j}\left(-t\langle U_i, V_j \rangle + \frac{b}{t}\right)\right) \leq \mathsf{m}^*.$$

Suppose, for the sake of contradiction, that for some $b, t$, we have that

$$\min\left(\min_i\left(\langle U_i, V_i \rangle - \frac{b}{t}\right), \min_{i \neq j}\left(-t\langle U_i, V_j \rangle + \frac{b}{t}\right)\right) = \mathsf{m}' > \mathsf{m}^*.$$

Then,

$$\min_i \langle U_i, V_i \rangle - \max_{i \neq j} \langle U_i, V_j \rangle \geq \frac{b}{t} + \mathsf{m}' - (\frac{b}{t} - \mathsf{m}') = 2\mathsf{m}' > 2\mathsf{m}^*,$$

which is a contradiction with the definition of $\mathsf{m}^*$. $\qquad\square$

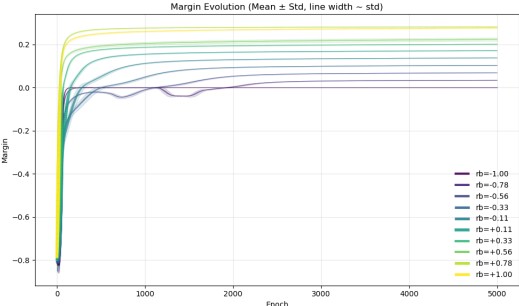

Figure 10: Evolution of margins when training with different fixed relative biases, average over 100 iterations.

## A.2 Robustness of Nearest-Neighbor Retrieval: Proof of Proposition 1

Suppose that

$$\mathop{\mathbb{E}}_{(a_j)_{j=1}^{B}\sim[N]^{\times k}}\Big[\sum_{j=1}^{B}\log\Big(1+\exp(-t\langle U_{a_j},V_{a_j}\rangle+b)\Big)$$
$$+\sum_{i\neq j}\log\Big(1+\exp(t\langle U_{a_i},V_{a_j}\rangle-b)\Big)\Big]\leq\xi\log 2.$$

holds. Again,as the function $x\longrightarrow\log(1+e^x)$ is non-negative, this implies that for some $0\leq x\leq\xi$,

$$x\log 2=\mathop{\mathbb{E}}_{(a_j)_{j=1}^{B}\sim[N]^{\times k}}\Big[\sum_{j=1}^{B}\log\Big(1+\exp(-t\langle U_{a_j},V_{a_j}\rangle+b)\Big)\Big]$$
$$=B\times\mathop{\mathbb{E}}_{j\sim\mathsf{unif}([N])}\log\Big(1+\exp(-t\langle U_j,V_j\rangle+b)\Big).$$

However, whenever $t\langle U_j,V_j\rangle-b\leq 0$, then $\log\Big(1+\exp(-t\langle U_j,V_j\rangle+b)\Big)\geq\log 2$. By Markov's inequality, it follows that

$$\mathop{\mathbb{P}}_{j\sim\mathsf{unif}([N])}[t\langle U_j,V_j\rangle-b\leq 0]\leq\frac{x}{B}.$$

Hence, for all but at most a $\frac{x}{B}$ fraction of the data indices $j$, it follows that $\langle U_j,V_j\rangle>b/t$.

In the exact same way, for some $y\geq 0$ such that $x+y\leq\xi$. it follows that

$$\mathop{\mathbb{P}}_{i,j\sim[N]^{\times 2}}[t\langle U_j,V_j\rangle-b\geq 0]\leq\frac{y}{B(B-1)}.$$

Note that for every fixed $i$, if $\mathbb{P}_{j\sim\mathsf{unif}([N])|_{j\neq i}}[t\langle U_j,V_j\rangle-b>0]>0$, then $\mathbb{P}_{j\sim\mathsf{unif}([N])|_{j\neq i}}[t\langle U_j,V_j\rangle-b>0]\geq 1/(N-1)$. Hence, one can similarly argue that for all but at most a $\frac{y(N-1)}{B(B-1)}$ fraction of the data indices $i$, it follows that $\langle U_i,V_j\rangle<b/t$ for all $j$. Thus, for at least a

$$1-\frac{x}{B}-\frac{y(N-1)}{B(B-1)}$$

fraction of the indices $i$, it follows that for any $j\neq i$,

$$\langle U_i,V_i\rangle>b/t>\langle U_i,V_j\rangle.$$

Clearly, for these indices, nearest neighbor search succeeds. Optimizing over $0\leq x,0\leq y,x+y\leq\xi$, we reach the conclusion.

## A.3 Triplet Loss

In the context of synchronizing embeddings, the triplet loss function [SKP15] with hyperparameter margin $\alpha$ takes form

$$\mathcal{L}^{\mathsf{Triplet}}(\{(U_i,V_i)\}_{i=1}^{N};\alpha)=\sum_{i\neq j}\max(\|U_i-V_i\|_2^2-\|U_i-V_j\|_2^2+\alpha,0). \tag{19}$$

**Observation 4.** *Suppose that $\{(U_i, V_i)\}_{i=1}^N$ is a $(\mathsf{m}, \mathsf{b_{rel}})$-Constellation. Then, for any $\alpha \leq 4\mathsf{m}$, it is also the case that*

$$\mathcal{L}^{\mathsf{Triplet}}\big(\{(U_i, V_i)\}_{i=1}^N; \alpha\big) = 0.$$

*Proof.* Suppose that $\{(U_i, V_i)\}_{i=1}^N$ is a $(\mathsf{m}, \mathsf{b_{rel}})$-Constellation. Then, for any $i, \neq j$, we have that

$$
\begin{aligned}
&\|U_i - V_i\|_2^2 - \|U_i - V_j\|_2^2 \\
&= 2 - 2\langle U_i, V_i \rangle - (2 - 2\langle U_i, V_j \rangle) \\
&= 2(\langle U_i, V_j \rangle - \langle U_i, V_i \rangle) \\
&\leq 2(-\mathsf{m} + \mathsf{b_{rel}} - \mathsf{m} - \mathsf{b_{rel}}) = -4\mathsf{m}.
\end{aligned}
$$

This finishes the proof. $\qquad\square$

# B  Proof of Theorem 3.5: Dimension vs Size tradeoff

*Proof of Theorem 3.5.* Let

$$H \sim \mathrm{Unif}(S^{d-1}), \qquad C(H) = \{\, i : \langle c_i, H \rangle > \delta \}, \quad N' = |C(H)|,$$

where $c_i = (U_i + V_i)/2$ and $\delta \in (0, 1)$ will be chosen later.

$$\|c_i\|^2 = \frac{1 + \langle U_i, V_i \rangle}{2} \in \left[\tfrac{1+\mathsf{m}+\mathsf{b_{rel}}}{2}, 1\right], \text{ so } \|c_i\| \geq \sqrt{(1 + \mathsf{m} + \mathsf{b_{rel}})/2}$$

Then the inner product satisfies

$$\Pr\big[\langle c_i, H \rangle > \delta\big] = \Gamma_d\Big(\tfrac{\delta}{\|c_i\|}\Big) \ \geq \ \Gamma_d\Big(\delta\sqrt{\tfrac{2}{1+\mathsf{m}+\mathsf{b_{rel}}}}\Big),$$

where $\Gamma_d(x) = \Pr[H_1 > x]$ is strictly decreasing in $x$. By linearity of expectation,

$$\mathbb{E}[N'] \ \geq \ N\,\Gamma_d\Big(\delta\sqrt{\tfrac{2}{1+\mathsf{m}+\mathsf{b_{rel}}}}\Big),$$

so there exists a realization of $H$ with

$$N' \ \geq \ N\,\Gamma_d\Big(\delta\sqrt{\tfrac{2}{1+\mathsf{m}+\mathsf{b_{rel}}}}\Big). \tag{20}$$

Define for the index set $C = C(H)$ the sums

$$U_C = \sum_{i \in C} U_i, \quad V_C = \sum_{i \in C} V_i, \quad x_i = U_i - V_i, \quad x_C = \sum_{i \in C} x_i,$$

and set

$$A_C = \sum_{i \in C} \langle U_i, V_i \rangle, \quad B_C = \sum_{\substack{i,j \in C \\ i \neq j}} \langle U_i, V_j \rangle, \quad \xi_C = \frac{1}{N'}\Big(\sum_{i \in C} \|x_i\|^2\Big) - \|\tfrac{1}{N'} x_C\|^2 \geq 0.$$

A direct expansion shows

$$\|U_C + V_C\|^2 = N'^2(2 - \xi_C) \ - \ 2(N' - 2)\,A_C \ + \ 4\,B_C. \tag{21}$$

On the other hand,

$$\Big\|\sum_{i \in C} c_i\Big\| \ = \ \frac{1}{2}\|U_C + V_C\| \ \geq \ \sum_{i \in C} \langle c_i, H \rangle \ > \ N'\,\delta,$$

so

$$\|U_C + V_C\|^2 > 4N'^2\delta^2. \tag{22}$$

Combining (21) and (22), and using $A_C \geq (\mathsf{m} + \mathsf{b_{rel}})N'$ and $B_C \leq (\mathsf{b_{rel}} - \mathsf{m})N'(N' - 1)$, yields

$$4N'^2\delta^2 \ \leq \ N'^2(2 - \xi_C) \ - 2(N' - 2)\,(\mathsf{m} + \mathsf{b_{rel}})N' \ + 4\big[(\mathsf{b_{rel}} - \mathsf{m})\,N'(N' - 1)\big]$$

Which means $N'\big(3\mathsf{m} - 1 + 2\delta^2 - \mathsf{b}_{\mathsf{rel}}\big) \leq 4\mathsf{m}$. Where in the last reduction we dropped the $\xi_C \geq 0$ term. Hence, whenever $2\delta^2 > 1 - 3\mathsf{m} + \mathsf{b}_{\mathsf{rel}}$,

$$N' \leq \frac{4\mathsf{m}}{2\delta^2 - (1 - 3\mathsf{m} + \mathsf{b}_{\mathsf{rel}})}. \tag{23}$$

Combine (20) and (23) to get

$$N \leq \frac{4\mathsf{m}}{2\delta^2 - (1 - 3\mathsf{m} + \mathsf{b}_{\mathsf{rel}})} \, \Gamma_d\Big(\delta\sqrt{\tfrac{2}{1+\mathsf{m}+\mathsf{b}_{\mathsf{rel}}}}\Big)^{-1}.$$

Recalling Shannon's asymptotic lower-bound $\Gamma_d(\cos\theta) = \exp\{d\log\sin\theta + o_\theta(d)\}$ [Sha59, (11)] with $\cos\theta = \delta\sqrt{2/(1+m+\mathsf{b}_{\mathsf{rel}})}$ and corresponding $\sin\theta = \sqrt{1 - \cos^2\theta} = \sqrt{1 - \frac{1-3m+\mathsf{b}_{\mathsf{rel}}}{1+m+\mathsf{b}_{\mathsf{rel}}}}$, the bound is optimized by choosing $\delta = \sqrt{\frac{1-3m+\mathsf{b}_{\mathsf{rel}}}{2}}$. This results in the claimed bound

$$N \leq \exp(-d\log\sin\theta + o_\theta(d)). \qquad \square$$

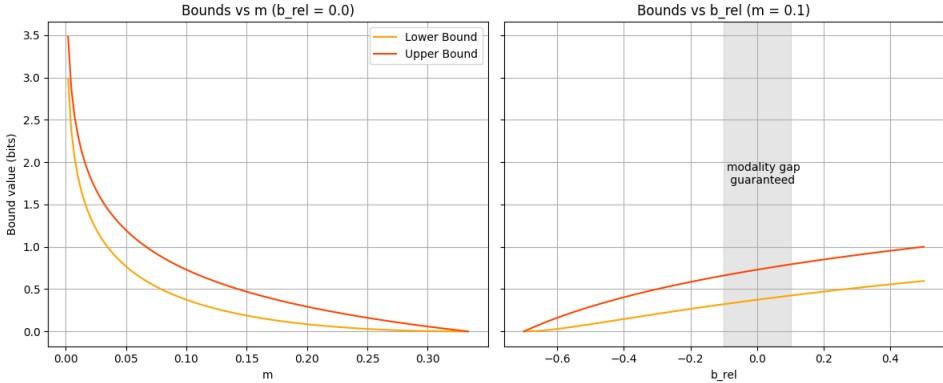

Figure 11: Upper and Lower Bounds from Theorem 3.5 and Theorem 3.3 for fixed $\mathsf{b}_{\mathsf{rel}} = 0$, $m = 0.1$.

## C  Omitted Proofs from Section 3.3: Combinatorics of The Modality Gap

Here, we analyze configurations $\{(U_i, V_i)\}_{i=1}^N \in (\mathbb{S}^{d-1} \times \mathbb{S}^{d-1})^{\times N}$ with the following property:

$$\begin{aligned} &\langle U_i, V_i \rangle > 0 \ \forall i, \\ &\langle U_i, V_j \rangle < 0 \ \forall i \neq j. \end{aligned} \tag{24}$$

Ultimately, we aim to prove Theorem 3.6. We also prove several other facts on the way. Our proofs are based on simple facts from convex geometry which we introduce now. One can find more, for example, in the excellent book [BMS12].

### C.1  Preliminaries from Convex Geometry

A set $K \subseteq \mathbb{R}^d$ is called *convex* if for any $\alpha \in [0, 1]$ and any $p, q \in K$, it is also the case that $\alpha p + (1 - \alpha)q \in K$. In particular, for any points $p_1, p_2, \ldots, p_k \in \mathbb{R}^d$, the following two sets are convex. The *convex hull* defined by

$$\mathsf{conv}(p_1, p_2, \ldots, p_n) \coloneqq \Big\{ \alpha_1 p_1, \alpha_2 p_2 + \cdots + \alpha_k p_k \ : \ \alpha_i \geq 0 \quad \forall i \text{ and } \sum_{i=1}^n \alpha_i = 1 \Big\}$$

and the *convex cone* defined by

$$\mathsf{cone}(p_1, p_2, \ldots, p_n) \coloneqq \Big\{ \alpha_1 p_1, \alpha_2 p_2 + \cdots + \alpha_k p_k \ : \ \alpha_i \geq 0 \quad \forall i \text{ and } \sum_{i=1}^n \alpha_i \leq 1 \Big\}$$

$$= \mathsf{conv}(p_1, p_2, \ldots, p_n, 0).$$

We also introduce the dual cone. For a set $S \subseteq \mathbb{R}^n$, the *dual cone* is given by

$$\mathrm{dualcone}(S) := \{v \in \mathbb{R}^n \; : \; \langle v, x \rangle \geq 0 \quad \forall x \in S\}.$$

We will use the following classic theorems from convex geometry.

**Theorem C.1** (Helly [Hel23]). *Let $X_1, X_2, \ldots, X_n$ be a finite collection of convex sets in $\mathbb{R}^d$. If the intersection of every $d + 1$ of these sets is nonempty, then the intersection of all the sets is nonempty. Formally,*

$$\text{If } \bigcap_{i \in I} X_i \neq \emptyset \quad \text{for all } I \subset \{1, 2, \ldots, n\} \text{ with } |I| = d + 1, \text{then } \bigcap_{i=1}^{n} X_i \neq \emptyset.$$

**Theorem C.2** (Carathéodory [Car11, Ste13]). *Let $A \subseteq \mathbb{R}^d$. If $\mathbf{x} \in conv(A)$, then there exists a set $B \subseteq A$ such that $|B| \leq d + 1$ and $\mathbf{x} \in \mathrm{conv}(B)$.*

**Theorem C.3** (Hyperplane Separation Theorem [Min10]). *Let $X$ and $Y$ be two nonempty, disjoint convex sets in $\mathbb{R}^d$. Then there exists a nonzero vector $a \in \mathbb{R}^d$ and a scalar $b$ such that*

$$\langle a, x \rangle \leq b \quad \text{for all } x \in X,$$
$$\langle a, y \rangle \geq b \quad \text{for all } y \in Y.$$

## C.2 Combinatorics of Modality Gap

**Proposition 2.** *If (24) hold and $N \geq d + 2$, then there exists some $h \in \mathbb{S}^{d-1}$ such that $\langle h, U_i \rangle > 0$ for all $i$.*

*Proof.* For each $i = 1, \ldots, N$, define the open half-space

$$H_i = \{\, x \in \mathbb{R}^d : \langle U_i, x \rangle > 0\}.$$

Each $H_i$ is convex. We first show that any subcollection of $d + 1$ of these half-spaces has nonempty intersection. Indeed, pick distinct indices $i_1, \ldots, i_{d+1}$; since $N \geq d + 2$, there is an index $j \notin \{i_1, \ldots, i_{d+1}\}$. By (24), for each $k = 1, \ldots, d + 1$,

$$\langle U_{i_k}, V_j \rangle < 0 \,, \text{ so } \langle U_{i_k}, -V_j \rangle > 0,$$

and $-V_j \in \bigcap_{k=1}^{d+1} H_{i_k} \neq \varnothing$.

Since every $d + 1$ of the $H_i$ intersect and $N \geq d + 2$, Helly's theorem implies

$$\bigcap_{i=1}^{N} H_i \neq \varnothing.$$

Choose any $h_0 \in \bigcap_{i=1}^{N} H_i$. Then $\langle h_0, U_i \rangle > 0$ for all $i$. Setting $h = h_0/\|h_0\| \in \mathbb{S}$ preserves these strict inequalities. $\qquad\square$

**Proposition 3.** *If (24) hold and $N \geq d + 2$, then there exists some $h \in \mathbb{R}^d$ such that $\langle h, U_i \rangle > 0$ for all $i$ and $h \in \mathrm{conv}(U_1, U_2, \ldots, U_N)$.*

*Proof.* Let $C := \mathrm{conv}(U_1, \ldots, U_N) \subset \mathbb{R}^d$, a compact convex set. Take the unit vector $h$ given by Proposition 2 and denote by $h'$ its projection onto $C$. This implies the fact that that the hyperplane

$$H := \big\{x \in \mathbb{R}^d : \langle h - h', \, x - h' \rangle = 0\big\}$$

*supports* the set $C$ at the point $h'$: all points of $C$ (hence each $U_i$) lie in the closed half-space

$$H^- := \big\{x \in \mathbb{R}^d : \langle h - h', \, x - h' \rangle \leq 0\big\}, \tag{25}$$

whereas $h$ itself belongs to the opposite open half-space $H^+ := \{x : \langle h - h', x - h' \rangle > 0\}$. Choose an orthonormal basis $\{e_1, \ldots, e_d\}$ with

$$e_1 := \frac{h - h'}{\|h - h'\|}.$$

In these coordinates

$$h = h' + \alpha e_1, \quad \alpha := \|h - h'\| > 0, \qquad h' = 0 \cdot e_1 + h'_\perp,$$

while every $U_i$ has a decomposition $U_i = U_{i,1}\, e_1 + U_{i,\perp}$ with $U_{i,1} \leq 0$.

For each $i$,

$$\langle h', U_i \rangle - \langle h, U_i \rangle = \langle h' - h,\, U_i \rangle = -\alpha\, U_{i,1} \ \geq\ 0.$$

Because $\langle h, U_i \rangle > 0$, we conclude that

$$\langle h', U_i \rangle > 0 \qquad (i = 1, \ldots, N). \tag{26}$$

$\square$

*Proof of Theorem 3.6.* Let $h$ be the vector provided by Proposition 3, so $h \in \text{conv}(U_1, \ldots, U_N)$ and $\langle h, U_i \rangle > 0$ for every $i$. Set the cone

$$C' := \text{cone}\{U_1, \ldots, U_N\} = \Big\{ \sum_{i=1}^{N} a_i U_i \ :\ a_i \geq 0 \Big\}.$$

Because each $U_i$ has positive dot product with $h$, define the affine hyperplane

$$H := \big\{ x \in \mathbb{R}^d : \langle x, h \rangle = 1 \big\}.$$

Every ray $\{\lambda U_i : \lambda > 0\}$ meets $H$ once, namely at

$$Q_i := \frac{1}{\langle U_i, h \rangle}\, U_i \ \in H.$$

Consequently

$$C' \cap H \ = \ \text{conv}\{Q_1, \ldots, Q_N\}. \tag{27}$$

The the vector $h^\dagger$ in $H$ which is parallel to $h$. $\langle h^\dagger, h \rangle = 1$, so $h^\dagger$ is $h$ rescaled by a positive scalar. Since $h^\dagger$ also lies in $\text{conv}(U_1, \ldots, U_N) \subset C'$, we have $h^\dagger \in C' \cap H$. The hyperplane $H$ is $(d-1)$–dimensional. By Carathéodory's theorem in $\mathbb{R}^{d-1}$, there is a subset $S \subset \{1, \ldots, N\}$ with $|S| \leq d$ and weights $\lambda_i \geq 0$, $\sum_{i \in S} \lambda_i = 1$, such that

$$h^\dagger \ = \ \sum_{i \in S} \lambda_i\, Q_i \ = \ \sum_{i \in S} \lambda_i\, \frac{U_i}{\langle U_i, h \rangle}. \tag{28}$$

Fix $k \notin S$. Using Theorem C.2 and (24),

$$\langle h^\dagger, V_k \rangle = \sum_{i \in S} \lambda_i\, \frac{\langle U_i, V_k \rangle}{\langle U_i, h \rangle} < 0,$$

because each numerator $\langle U_i, V_k \rangle$ is negative and each denominator $\langle U_i, h \rangle$ is positive. Hence $\langle h^\dagger, V_k \rangle < 0$ for every $k \notin S$. Only the (at most) $d$ indices in $S$ may give a non–negative value. Note that $h^\dagger$ is already on the unit circle. $\square$

Now we prove that there is a construction for which this bound is almost tight and we can separate all but at least $d - 1$ vectors with a hyperplane.

**Construction 3** (Tightness of Theorem 3.6)**.** *There exists a set of vectors $\{(U_i, V_i)\}_{i=1}^{N}$ such that $\langle U_i, V_i \rangle > 0$ for each $i$, $\langle U_i, V_j \rangle < 0$ for each $i \neq j$, and for any $h \in \mathbb{S}^{d-1}$, at least for $d - 1$ values of $i$, it holds that $\langle h, U_i \rangle$ and $\langle h, V_i \rangle$ have the same sign.*

*Proof.* For the construction for $d = 3$, note that we can take two parallel $k$-gons equally far from the equator of the sphere such that the zeniths of their points for one of them is $\pi/4 + \delta$ and for the other one is $3/4\pi - \delta$ and by taking $\delta \to 0$ we can ensure that the dot product between corresponding pairs is positive and the dot product between non-matching pairs is negative. Now note that by taking $k$ sufficiently large and the $\delta$ sufficiently small. The intersection of that configuration with the wedge with dihedral angle $\alpha$ contains at least $N - 2$ points from one of the $k$-gons (the $U$'s) and $N - 2$ points from the other (the $V$'s) for any value of $\alpha$. See Fig. 12.

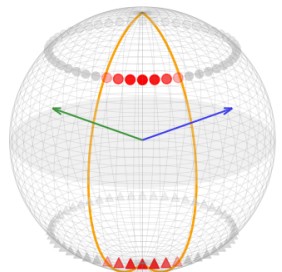

Figure 12: Construction in 3D.

Label those as

$$U_1, \ldots, U_{N-2} \quad \text{and} \quad V_1, \ldots, V_{N-2}.$$

Finally place two more pairs on the equator:

$$U_{N-1} = V_{N-1}, \quad U_N = V_N$$

chosen so that $\langle U_{N-1}, U_N \rangle < 0$ and both make negative dot-products with each wedge boundary ray. This completes the $d = 3$ example.

For the general case ($d > 3$). Let $\omega_1, \ldots, \omega_{d+1}$ be the vertices of a regular simplex in $\mathbb{R}^d$. Set

$$U_i = V_i = \omega_i, \quad i = 1, \ldots, d-3.$$

These occupy a $(d-3)$–dimensional subspace. In the orthogonal complement (which is 3–dimensional), embed the $d = 3$ construction, obtaining pairs $(U_{d-2}, V_{d-2}), \ldots, (U_N, V_N)$. Finally, pick a small $\varepsilon > 0$ and renormalize:

$$U_i' = \varepsilon\, \omega_{d-2} \ + \ \sqrt{1 - \varepsilon^2}\, U_i, \quad V_i' = \varepsilon\, \omega_{d-2} \ + \ \sqrt{1 - \varepsilon^2}\, V_i, \quad i = d-2, \ldots, N.$$

For $\varepsilon$ sufficiently small, all the required dot-product signs are preserved, and since the configuration was orthogonal to $U_i$ for $i = 1, 2, \ldots, d-3$,

$$\langle U_i, V_j \rangle = -\epsilon\frac{1}{d} < 0, \forall i = 1, 2, \ldots, d-3, j = d-2, d-1, \ldots, N,$$

as needed. □

**Proposition 4.** *If* (24) *hold then* $U_i \notin \text{cone}(\{U_j\}_{j \neq i})$ *for any* $i$.

*Proof.* Suppose, to the contrary, that for some fixed $i$ there exist scalars $a_j \geq 0$ for $j \neq i$ such that

$$U_i \ = \ \sum_{j \neq i} a_j\, U_j.$$

Taking the inner product with $V_i$ gives

$$\langle V_i, U_i \rangle \ = \ \sum_{j \neq i} a_j\, \langle V_i, U_j \rangle.$$

Since by (24) we have $\langle V_i, U_j \rangle < 0$ for all $j \neq i$ and each $a_j \geq 0$, the right–hand side is nonpositive. But the left–hand side is strictly positive, a contradiction. Therefore $U_i \notin \text{cone}(\{U_j\}_{j \neq i})$. □

**Proposition 5.** *If* $d = 2$ *and* $N \geq 4$*, there does not exist a configuration satisfying* (24).

*Proof.* Because $N \geq d + 2$, Proposition 3 gives a unit vector $h$ with $\langle h, U_i \rangle > 0$ for every $i$. Rotate so $h = (1,0)$; all $U_i$ now lie in the open right half-plane $x > 0$. Write their polar angles in $(-\frac{\pi}{2}, \frac{\pi}{2})$ as

$$-\frac{\pi}{2} < \theta_1 < \theta_2 < \cdots < \theta_N < \frac{\pi}{2}.$$

Now note that $U_2 \in \mathsf{cone}(U_1, U_3)$ which is a contradiction by Proposition 4. Therefore configuration (24) cannot exist when $d = 2$ and $N \geq 4$. $\square$

# D  Further Experiments and Experimental Details

For the experiments in Appendix D.1, we used a single A100 GPU. All other experiments are done on a standard CPU and take at most several minutes.

## D.1  Experiments on ImageNet

In Figures 1 and 3, we performed experiments on real data with the SigLIP implementation. While a next generation vision-language encoder was introduced with the SigLIP 2 paper [TGW+25a], we opted to use the original SigLIP model rather than SigLIP 2 because SigLIP 2's enhanced training recipe – incorporating auxiliary decoder, self-distillation, and masked-prediction losses – would confound our ability to isolate the impact of the core Sigmoid Contrastive Loss on the embeddings. The data we used is the validation dataset of ImageNet which contains 50000 captioned images with 1000 distinct captions. We used 8 trained models listed in Table 5 which can all be downloaded from Hugging Face.

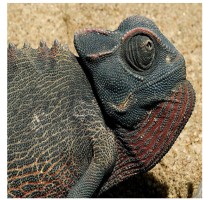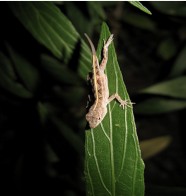

Figure 13: "African chameleon" on the right and "American chameleon" on the left from the ImageNet validation dataset. The B/16 model representation of the image of "American chameleon" was closer to the representation of the word African chameleon than that of American chameleon

We embedded all images and labels in the validation set using the B/16 model. We used PIL to resize all images to 224x24. In Figure 1, we show in red the inner products between wrong image-caption pairs and in blue between correct image-caption pairs.

As we point out, the inner product separation is nearly satisfied. There are some errors, but such are expected. For example, we discovered that the best matching image embeddings picture of the word "African chameleon" was "American chameleon". Both are species of chameleon and, hence, the images similar, such errors are to be expected in practice. For large models, the reported accuracy on ImageNet in [ZMKB23b] is 84.5%.

## D.2  Experiments with Locked Representation

In Figure 4, we performed experiments in which one modality is fixed. Namely, we first draw $\{U_i\}_{i=1}^N$ uniformly on the sphere and then fix them. Then, we try to synchronize with $\{V_i\}_{i=1}^N$ by running gradient descent on the respective loss function. Specifically, we have experiments on:

- *Fixed Low Temperature $t = 200$ and bias $b = 0$.* We fix $t = 200, b = 0$ and run Adam on $\{V_i\}_{i=1}^N$ for the loss $\mathcal{L}^{\mathsf{Sig}}(\{U_i\}_{i=1}^N, \{V_i\}_{i=1}^N; t, b)$ and initial learning rate 0.01.
- *Fixed High Temperature $t = 10$ and bias $b = 0$.* We fix $t = 10, b = 0$ and run Adam on $\{V_i\}_{i=1}^N$ for the loss $\mathcal{L}^{\mathsf{Sig}}(\{U_i\}_{i=1}^N, \{V_i\}_{i=1}^N); t, b$ and initial learning rate 0.01.
- *Trainable Temperature and Bias.* We initialize at $t = 10 = e^{t'}, b = 0$ and run Adam with on $\{V_i\}_{i=1}^N, t', b$ for the loss $\mathcal{L}^{\mathsf{Sig}}(\{U_i\}_{i=1}^N, \{V_i\}_{i=1}^N; e^{t'}, b)$ and initial learning rate 0.01. We note that all of our trainable experiments are with the parametrization $t = e^{t'}$ which ensures positive temperature as in [ZMKB23b].
- *Trainable Temperature and Relative Bias.* We initialize at $t = 10 = e^{t'}, b = 0$ and run Adam on $\{V_i\}_{i=1}^N, t', b_{\mathsf{rel}}$ for the loss $\mathcal{L}^{\mathsf{RB-Sig}}(\{U_i\}_{i=1}^N, \{V_i\}_{i=1}^N; e^{t'}, b_{\mathsf{rel}})$ and initial learning rate 0.01. We note that all of our trainable experiments are with the parametrization $t = e^{t'}$ which ensures positive temperature as in [ZMKB23b].

The specific experiment in Fig. 7 is for $d = 10, N = 100$. We also note that we did one more comparison, which is not reported in the main paper – with an explicit adapter from Figure 4. Namely:

- *Trainable Temperature and Relative Bias with Explicit Adapter.* We initialize at $t = 10 = e^{t'}, b = 0, \delta = \frac{e^x}{1+e^x}$ with $x = 1/2$ and run Adam on $\{V_i\}_{i=1}^N, t', \mathsf{b_{rel}}, x$ for the loss $\mathcal{L}^{\mathsf{RB-Sig}}(\{A^\delta_{\mathsf{locked}}(U_i)\}_{i=1}^N, \{A^\delta_{\mathsf{trainable}}(V_i)\}_{i=1}^N; e^{t'}, \mathsf{b_{rel}})$ and initial learning rate 0.01. Since the adapter is an invertible transformation on the representations, we reported the inner products both with the adapter and without it (that is, we invert by removing the last coordinate and dividing by $\delta$.)

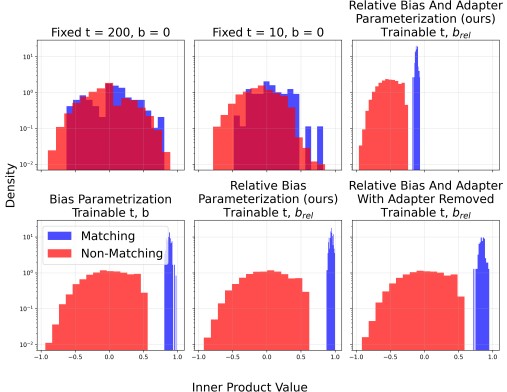 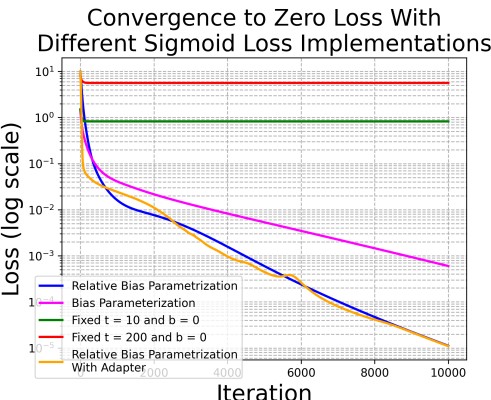

Figure 14: Inner-product separation and loss convergence under six sigmoid-loss parameterizations. *Left:* Log-density histograms of inner-product scores for matching (blue) versus non-matching (red) pairs, evaluated under fixed inverse temperature $t = 200, b = 0$, fixed $t = 10, b = 0$, trainable bias $b$, our relative-bias parameterization (trainable $\mathsf{b_{rel}}$), and the same two schemes with the adapter removed; only the trainable-bias models show clear separation. *Right:* Sigmoid-loss trajectories (log scale) over 10,000 iterations for the same six settings; only those variants that learn both bias and inverse temperature reach zero loss, and our relative-bias parameterization (with and without adapter) converges most rapidly.

We can overall see that the performance of $\mathcal{L}^{\mathsf{RB-Sig}}$ algorithm with an adapter and without is rather comparable and the inner product separations are similar. One difference to note is that the training with adapter seems less stable. Thus, we believe that in practice not using the adapter might be the better approach.

### D.3 Experiments with Multiple Modalities

In Figure 8, we performed experiments with $k = 4$ modalities. Namely, we synchronize $\{(U_i^{(1)}, U_i^{(2)}, U_i^{(3)}, U_i^{(4)})\}_{i=1}^N$ by running gradient descent on the sums of all pairwise loss functions between the 4 modalities. Specifically, we have experiments on:

- *Fixed Low Temperature $t = 200$ and bias $b = 0$.* We fix $t = 200, b = 0$ and run Adam on $\{V_i\}_{i=1}^N$ for the loss $\mathcal{L}^{\mathsf{Sig}}(\{U_i\}_{i=1}^N, \{V_i\}_{i=1}^N; t, b)$ and initial learning rate 0.01.

- *Fixed High Temperature $t = 10$ and bias $b = 0$.* We fix $t = 10, b = 0$ and run Adam on $\{V_i\}_{i=1}^N$ for the loss $\mathcal{L}^{\mathsf{Sig}}(\{U_i\}_{i=1}^N, \{V_i\}_{i=1}^N); t, b$ and initial learning rate 0.01.

- *Trainable Temperature and Bias.* We initialize at $t = 10 = e^{t'}, b = 0$ and run Adam with on $\{V_i\}_{i=1}^N, t', b$ for the loss $\mathcal{L}^{\mathsf{Sig}}(\{U_i\}_{i=1}^N, \{V_i\}_{i=1}^N; e^{t'}, b)$ and initial learning rate 0.01. We note that all of our trainable experiments are with the parametrization $t = e^{t'}$ which ensures positive temperature as in [ZMKB23b].

- *Trainable Temperature and Relative Bias.* We initialize at $t = 10 = e^{t'}, b = 0$ and run Adam on $\{V_i\}_{i=1}^N, t', \mathsf{b_{rel}}$ for the loss $\mathcal{L}^{\mathsf{RB-Sig}}(\{U_i\}_{i=1}^N, \{V_i\}_{i=1}^N; e^{t'}, \mathsf{b_{rel}})$ and initial learning rate

0.01. We note that all of our trainable experiments are with the parametrization $t = e^{t'}$ which ensures positive temperature as in [ZMKB23b].

The specific experiment in Fig. 8 is for $d = 10, N = 100$.

We ran additional experiments to investigate how increasing the number of modalities $k$ affects the final separation margin. With trainable temperature and relative bias, we observe that the margin generally increases as we synchronize more modalities, as summarized in Table 1. This suggests that training with more modalities may lead to more robust representations, as a larger margin implies better separation between matching and non-matching pairs.

Table 1: Final margin as a function of the number of modalities being synchronized. The experiment was run with $N = 100$ and $d = 10$.

| Number of Modalities | Final Margin |
|:---:|:---:|
| 2 | 0.471241 |
| 4 | 0.427528 |
| 6 | 0.472571 |
| 8 | 0.595576 |
| 14 | 0.610853 |
| 20 | 0.611314 |

### D.4 Bias Parameterization Leads to Zero Relative Bias

Finally, we do experiments to show that training with $\mathcal{L}^{\mathsf{Sig}}$ leads to near zero relative bias, as in [ZMKB23a]. We compare with $\mathcal{L}^{\mathsf{RB-Sig}}$. Concretely, we run experiments with $N = 100$ points $\{(U_i, V_i)\}_{i=1}^N$ initialized at random and run Adam on $\mathcal{L}^{\mathsf{Sig}}(\{(U_i, V_i)\}_{i=1}^N; t, b)$, respectively $\mathcal{L}^{\mathsf{RB-Sig}}(\{(U_i, V_i)\}_{i=1}^N; t, \mathsf{b}_{\mathsf{rel}})$, for 10000 epochs starting at $t = 10$ and varying biases.

We compare the evolution of relative biases, inverse temperature, loss function, and margins of the final configuration.

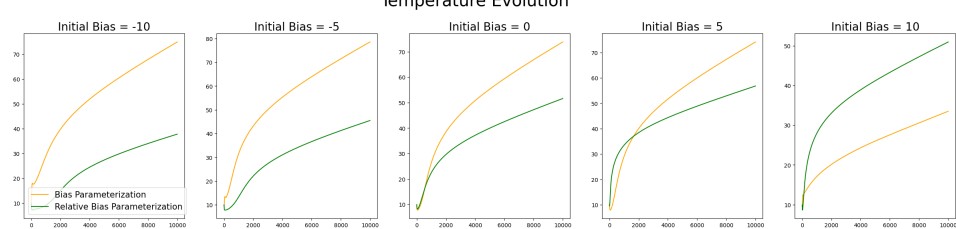

Figure 15: Evolution of the inverse temperature parameter during the training process.

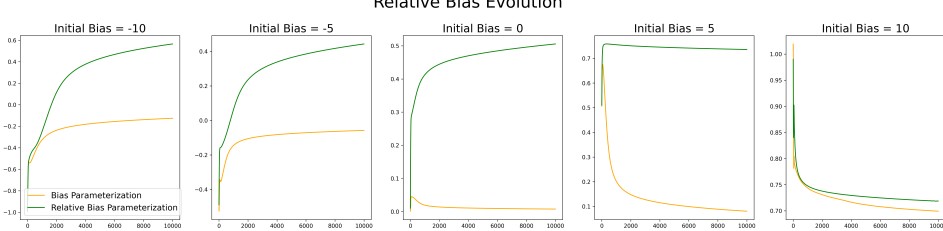

Figure 16: Relative bias is in general smaller when training with the $\mathcal{L}^{\mathsf{Sig}}$ parameterization. In general, it converges to zero and is significantly smaller than the relative bias of the $\mathcal{L}^{\mathsf{RB-Sig}}$.

Finally, we also compare the margins. The reason is that as we know from Theorem 3.4, there is an important relationship between relative bias and margin. The fact that embeddings trained with $\mathcal{L}^{\mathsf{RB-Sig}}$ have a larger relative bias also impacts the margin.

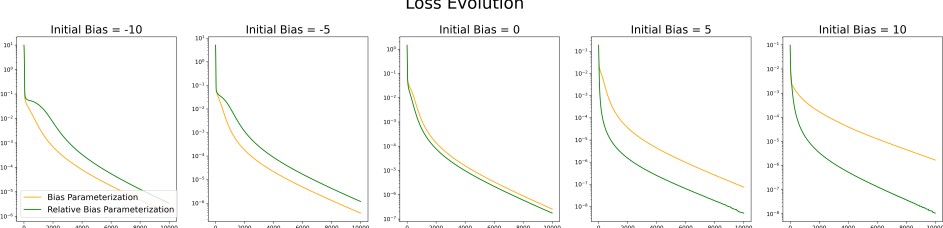

Figure 17: In general, the loss converges faster to zero when trained with the $\mathcal{L}^{\text{RB}-\text{Sig}}$ parameterization than when trained with $\mathcal{L}^{\text{Sig}}$.

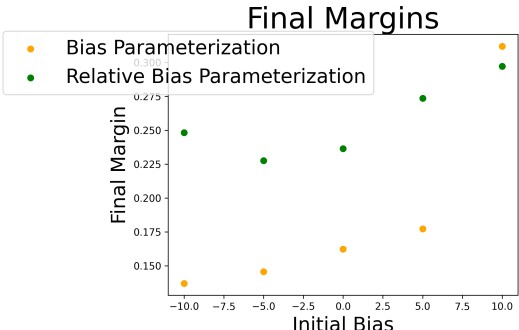

Figure 18: In general, the margin is much larger for representations trained with the $\mathcal{L}^{\text{RB}-\text{Sig}}$ parameterization. As we know from 1, this means that they are more robust on retrieval tasks.

### D.5 Initializing Fixed Relative Bias

We verify this with an experiment where we initialize representations uniformly at random, fix the relative bias $b_{\text{rel}}$, and train the representations and inverse temperature $t$ using Adam. As shown in Table 2, choosing $b_{\text{rel}} \approx 0.7$ yields the largest final margin, while other choices result in smaller margins. This confirms that the relative bias parameter can effectively steer the optimization towards configurations with desirable properties.

### D.6 Initializing Learnable Temperature and Relative Bias.

We investigated the effect of initial temperature $t$ and relative bias $b_{\text{rel}}$ on the final margin. We ran a hyperparameter search and found that the final margin is best for a small initial temperature ($t \leq 3$) or an intermediate temperature ($t \approx 10$) with a relatively large initial relative bias ($b_{\text{rel}} \approx 0.6$). The results, summarized in Table 3, show that while the optimization is robust to a range of initializations, a poor choice (e.g., high initial $t$ and low $b_{\text{rel}}$) can lead to suboptimal final representations with a small or even negative margin.

## E    Connection to Linear Representation Hypothesis Across Modalities

It has been observed by many authors that modern dense embedding spaces acquire correspondence between linear-algebraic operations and real-world concepts. This has been immortalized as "King - Man + Woman ≈ Queen" in word2vec [MCCD13] and is also observed in modern LLMs as well [PCV24, TCM⁺24]. Curiously, we find that contrastive pretraining with sigmoid loss also leads to a special case of LRH: there emerges a direction $\bar{x}$ such that adding it to an image embedding (almost) recovers the embedding of a matching text caption. Indeed, looking at the optimal embeddings in Fig. 5 we can see that $U_i - V_i$ does not depend on $i$, which we take as a manifestation of LRH in this context (the concept being "shift text to image" or more generally one modality to another). Furthermore, both our upper and lower bounds on the cardinality of the embeddings in Section 3.2 require Cross-Modality-LRH satisfying configurations to be tight.

Table 2: Final margin and loss for different fixed values of relative bias $b_{rel}$. Training is performed on the representations and inverse temperature $t$. The largest margin is achieved for $b_{rel} \approx 0.7$.

| Fixed Relative Bias | Final Temperature | Achieved Margin | Final Loss |
|---|---|---|---|
| -1.00 | 6.961601 | -0.000001 | 0.693150 |
| -0.90 | 56.188858 | -0.000000 | 0.009245 |
| -0.80 | 23.300198 | -0.000000 | 0.014105 |
| -0.70 | 162.664841 | 0.092437 | 0.000005 |
| -0.60 | 125.656731 | 0.122326 | 0.000003 |
| -0.50 | 104.095329 | 0.152893 | 0.000003 |
| -0.40 | 90.788620 | 0.182992 | 0.000002 |
| -0.30 | 81.079422 | 0.213618 | 0.000001 |
| -0.20 | 75.061546 | 0.242438 | 0.000001 |
| -0.10 | 71.254242 | 0.273600 | 0.000000 |
| 0.00 | 69.018265 | 0.301340 | 0.000000 |
| 0.10 | 69.534515 | 0.329022 | 0.000000 |
| 0.20 | 67.996437 | 0.353406 | 0.000000 |
| 0.30 | 61.796310 | 0.390087 | 0.000000 |
| 0.40 | 55.452553 | 0.430921 | 0.000000 |
| 0.50 | 48.406261 | 0.466707 | 0.000000 |
| 0.60 | 44.391388 | 0.498564 | 0.000000 |
| 0.70 | 42.265457 | 0.527834 | 0.000000 |
| 0.80 | 37.361767 | 0.539749 | 0.000001 |
| 0.90 | 33.167175 | 0.483351 | 0.000036 |
| 1.00 | 23.817665 | 0.513416 | 0.000693 |

Table 3: The final margin achieved for different initializations of temperature (Temp) and relative bias ($b_{rel}$). The best results (bolded) are obtained with low-to-intermediate temperature and high relative bias.

| Temp | -1.0 | -0.8 | -0.6 | -0.4 | -0.2 | 0.0 | 0.2 | 0.4 | 0.6 | 0.8 | 1.0 |
|---|---|---|---|---|---|---|---|---|---|---|---|
| 1 | 0.567 | 0.567 | 0.566 | 0.564 | 0.566 | 0.566 | 0.565 | 0.568 | 0.570 | **0.574** | 0.573 |
| 3 | 0.545 | 0.543 | 0.526 | 0.499 | 0.483 | 0.488 | 0.536 | 0.563 | 0.570 | **0.574** | 0.573 |
| 10 | 0.439 | 0.425 | 0.415 | 0.406 | 0.402 | 0.410 | 0.429 | 0.524 | 0.566 | 0.547 | 0.562 |
| 30 | 0.301 | 0.297 | 0.294 | 0.315 | 0.343 | -0.942 | -0.915 | -0.935 | -1.171 | -1.051 | -1.145 |
| 100 | -0.774 | -0.679 | -0.483 | -0.740 | -0.878 | -0.956 | -0.978 | -1.109 | -1.186 | -1.080 | -1.449 |

In the proof of Theorem 3.5 in Appendix B we defined the following quantity characterizing an arbitrary constellation

$$\xi = \frac{1}{N}\left(\sum_i \|x_i\|^2\right) - \|\bar{x}\|^2 \geq 0\,,$$

where $x_i = U_i - V_i$ and $\bar{x} = \frac{1}{N}\sum_i x_i$. (In the proof $\xi$ was defined for a carefully chosen sub-constellation). We note that the upper bound in that Theorem 3.5 could only possibly be tight if $\xi \approx 0$. In this section we will further show that $\xi$ can be used as a quantitative measure of the degree to which *Linear Representation Hypothesis* is satisfied.

First, let us establish that when $\xi$ is small (as $\xi \geq 0$ is used in the proof of Theorem 3.5 this means that the bounds in that proof are tight). More importantly, a small $\xi$ implies something significant about our representations: it suggests that the difference vector, $U_i - V_i$, is nearly identical for all indices $i$. Think of it this way: if you have two sets of learned representations, say $U_i$ for images and $V_i$ for their corresponding text descriptions, a small $\xi$ means that you can apply a consistent shift (a single vector) to all the $V_i$ vectors to transform them into their corresponding $U_i$ vectors. So, by shifting a text representation, you could get its corresponding image representation.

**Proposition 6.** *If $\xi = o(1)$ then $\frac{1}{N}\sum_{i=1}^{N} \|x_i - \bar{x}\|^2 = o(1)$ and all pairs of representations align in the sense that $U_i - V_i \approx U_j - V_j$ for all $i, j$. In particular, the U's are obtained from the V's by adding a vector $\bar{x}$, which thus serves as a concept shift.*

*Proof.* A simple algebra shows that $\xi$ has the following two equivalent expressions:

$$\xi = \frac{1}{N}\sum_{i=1}^{N}\|x_i - \bar{x}\|^2 = \frac{1}{2N^2}\sum_{i,j=1}^{N}\|x_i - x_j\|^2.$$

Thus, the statement "$\xi = o(1)$" is equivalent to

$$0 \le \frac{1}{N}\sum_{i=1}^{N}\|x_i - \bar{x}\|^2 \to 0,$$

which in turn implies that $U_i - V_i \approx \bar{x}$ simultaneously for all $i$. The argument for $U_i - V_i \approx U_j - V_j$ is similar. $\square$

**Corollary 2.** *If $\xi = 0$, then $U_i - V_i$ are all identical for all indices $i$.*

Indeed, in Fig. 19 when training directly the representations, we can observe the $\xi$ value converging to 0 for a range of different dimensions.

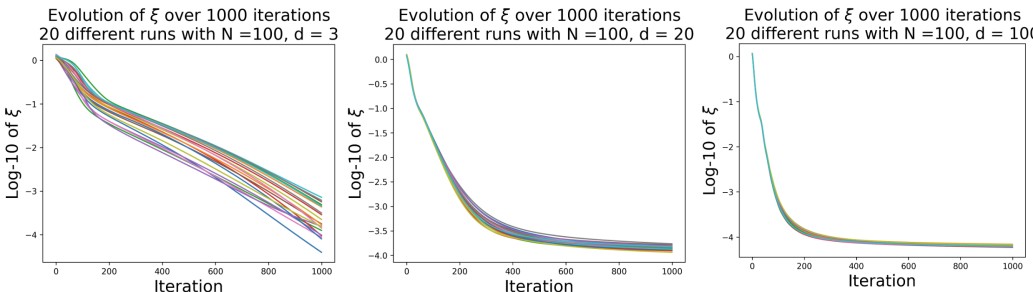

Figure 19: Convergence of the value of $\xi$ to zero during an experiment with $d = 10$ and $100$ $U_i, V_i$ pairs trained with SigLIP.

However, the value of $\xi$ for real models is far from 0 on the ImageNet validation dataset. Our intuition for this is that the dimension used $d \approx 1000$ is far from optimal, hence $\xi = 0$ is not required. It is an interesting open direction whether we can train models in lower dimension utilizing the fact that $\xi \longrightarrow 0$ in that case. For example, one can explicitly add $\xi$ in the loss function.

Table 4: $\xi$ for different SigLIP models in ImageNet validation. We plot respectively $\xi$, $\frac{1}{N}\left(\sum_i \|x_i\|^2\right)$ as mean of norms, $\|\bar{x}\|^2$ as norm of mean, $\frac{1}{N}\left(\sum_i \|U_i - V_{\pi(i)}\|^2\right)$ as random mean of norms where $\pi$ is a uniformly random permutation. We can see that the mean of norms is closer to the random mean of norms (i.e., random pairing of text-images, not corresponding to the ground truth) rather than the norm of means, which would imply $\xi \longrightarrow 0$.

| Model | $\xi$ | Mean of Norms | Norm of Mean | Random Mean of Norms |
|---|---|---|---|---|
| siglip-so400m-patch14-384 | 0.6086 | 1.7249 | 1.1162 | 2.0029 |
| siglip-base-patch16-224 | 0.5880 | 1.8100 | 1.2221 | 2.0609 |
| siglip-base-patch16-384 | 0.5908 | 1.8068 | 1.2160 | 2.0631 |
| siglip-large-patch16-256 | 0.5535 | 1.7955 | 1.2420 | 2.0711 |
| siglip-so400m-patch14-224 | 0.6207 | 1.7270 | 1.1063 | 2.0038 |
| siglip-base-patch16-256 | 0.5767 | 1.7991 | 1.2225 | 2.0588 |
| siglip-base-patch16-512 | 0.5908 | 1.8059 | 1.2151 | 2.0644 |
| siglip-large-patch16-384 | 0.5744 | 1.8084 | 1.2340 | 2.0762 |

Table 5: Margin and Relative bias corresponding to 5th-percentile positive and 95-th percentile negative pairs for different SigLIP models. We highlight that the two largest so400m models have a substantially different relative bias than the rest of the models. Likewise, the margin is perfectly correlated with the embedding dimension – bigger models have bigger margin.

| Model | 5% Positive Pairs | 95% Negative Pairs | Margin | Relative Bias | Dimension |
|---|---|---|---|---|---|
| siglip-so400m-patch14-384 | 0.0769 | 0.0486 | 0.0142 | 0.0627 | 1152 |
| siglip-so400m-patch14-224 | 0.0747 | 0.0483 | 0.0132 | 0.0615 | 1152 |
| siglip-large-patch16-256 | 0.0400 | 0.0151 | 0.0124 | 0.0276 | 1024 |
| siglip-large-patch16-384 | 0.0353 | 0.0120 | 0.0117 | 0.0237 | 1024 |
| siglip-base-patch16-512 | 0.0409 | 0.0170 | 0.0120 | 0.0289 | 768 |
| siglip-base-patch16-384 | 0.0408 | 0.0173 | 0.0118 | 0.0290 | 768 |
| siglip-base-patch16-256 | 0.0413 | 0.0200 | 0.0106 | 0.0306 | 768 |
| siglip-base-patch16-224 | 0.0383 | 0.0181 | 0.0101 | 0.0282 | 768 |

Table 6: Mean cosine similarities, margin, and relative bias for different SigLIP models. It is interesting to consider that the so400m are exactly on the boundary where the modality gap is guaranteed (all 8 models do satisfy the modality gap with zero misclassification error). The difference in margins is partly explained by dimensionality – larger dimensions correspond to larger margins. The Pearson correlation coefficient between dimension and margin is .948 and the Spearman coefficient is .926.

| Model | Mean Pos. Pairs | Mean Neg. Pairs | Margin | Relative Bias | Dimension |
|---|---|---|---|---|---|
| siglip-so400m-patch14-384 | 0.1376 | -0.0015 | 0.0695 | 0.0680 | 1152 |
| siglip-so400m-patch14-224 | 0.1365 | -0.0022 | 0.0694 | 0.0672 | 1152 |
| siglip-large-patch16-256 | 0.1023 | -0.0359 | 0.0691 | 0.0332 | 1024 |
| siglip-large-patch16-384 | 0.0958 | -0.0384 | 0.0671 | 0.0287 | 1024 |
| siglip-base-patch16-256 | 0.1004 | -0.0294 | 0.0649 | 0.0355 | 768 |
| siglip-base-patch16-512 | 0.0971 | -0.0322 | 0.0646 | 0.0324 | 768 |
| siglip-base-patch16-384 | 0.0966 | -0.0319 | 0.0642 | 0.0324 | 768 |
| siglip-base-patch16-224 | 0.0950 | -0.0305 | 0.0627 | 0.0322 | 768 |

