# OpenReview forum: "Global Minimizers of Sigmoid Contrastive Loss"
_NeurIPS.cc/2025/Conference — NeurIPS 2025 poster_

### Official Review · Reviewer_9MsS · 2025-06-21

**Clarity:** 3
**Significance:** 3
**Originality:** 3
**Rating:** 4
**Confidence:** 3

**Summary:**

This paper provides a theoretical analysis of representation synchronization under the sigmoid contrastive loss with trainable inverse temperature and bias, as used in recent SigLIP models. The authors introduce the novel concept of (m,rb)-Constellations, a class of configurations related to spherical codes and characterized by a margin and a relative bias. The paper leverages this formulation to explain the empirical success of SigLIP in retrieval tasks, interpret the presence of the modality gap, and identify the necessary dimension for producing high-quality representations. Additionally, the authors propose a reparameterization of the sigmoid loss using explicit relative bias, enabling practical improvements in scenarios such as synchronization with a locked encoder or across more than two modalities.

**Questions:**

In practice, over-eliminating the modality gap would not improve or even degrade the performance of the multimodal model. It would be helpful if the authors could provide a theoretical perspective on the optimal degree of separation between modalities in the context of the sigmoid contrastive loss. Is this gap also generalizable to InfoNCE contrastive loss ?


Can the authors analyze how the model's performance on standard benchmarks evolves as the sigmoid contrastive loss progressively approaches zero during training?

It would be valuable for the authors to offer insights into whether minimizing the sigmoid contrastive loss to zero might adversely affect the model’s generalization ability, e.g. zero-shot retrieval or under query distribution shifts.

**Ethical Concerns:**

["NO or VERY MINOR ethics concerns only"]

**Final Justification:**

The authors' rebuttal effectively addresses my concerns.

Therefore, I tend to maintain my score.

**Limitations:**

The authors have discussed the limitations in Section 4.

**Paper Formatting Concerns:**

The paper formatting looks good to me.

**Quality:**

3

**Strengths And Weaknesses:**

Strength

The paper provides the first characterization of global minimizers for sigmoid contrastive loss, with the number of data points N greatly exceeding the embedding dimension d.

The proposed reparameterization of the sigmoid loss with relative bias provides both theoretical insights and practical benefits, such as faster convergence and better performance in locked encoder scenarios.

Weakness

In practice, over-eliminating the modality gap would not improve or even degrade the performance of the multimodal model. It would be helpful if the authors could provide a theoretical perspective on the optimal degree of separation between modalities in the context of the sigmoid contrastive loss. Is this gap also generalizable to InfoNCE contrastive loss ?


Can the authors analyze how the model's performance on standard benchmarks evolves as the sigmoid contrastive loss progressively approaches zero during training?

It would be valuable for the authors to offer insights into whether minimizing the sigmoid contrastive loss to zero might adversely affect the model’s generalization ability, e.g. zero-shot retrieval or under query distribution shifts.

---

> ### Author Rebuttal · Authors · 2025-07-31
>
> > **Q1:** In practice, over-eliminating the modality gap would not improve or even degrade the performance of the multimodal model. It would be helpful if the authors could provide a theoretical perspective on the optimal degree of separation between modalities in the context of the sigmoid contrastive loss. Is this gap also generalizable to InfoNCE contrastive loss ?
>
> **А:** We thank the reviewer for suggesting the direction of theoretically understanding what the optimal modality gap is.
>
> Currently, our results are more qualitative and are aimed at justifying why multimodal models trained with sigmoid loss exhibit a modality gap. We do not, however, suggest that one should aim to eliminate the modality gap -- in fact, as written on page 3, "we leverage the modality gap to build a linear adapter which can be
> used towards synchronizing representations when one encoder is locked." Thus, we view the modality gap not necessarily as a drawback of multimodal models, but rather as a necessity for state of the art performance. Again, as explained on page 3, as "different modalities
> may contain different information it is only natural that
> they be represented in disjoint parts of the space."
>
> We note that in taking this perspective, we crucially depart from previous theoretical work such as "Understanding Contrastive Representation Learning through
> Alignment and Uniformity on the Hypersphere" by Tongzhou Wang and Phillip Isola. Prior work in the literature aims to show convergence to "aligned configurations". Such may not be optimal and are also not realistic. Both models trained with sigmoid loss (such as SigLIP) and models trained with InfoNCE loss (such as CLIP) exhibit a modality gap -- for CLIP, see Liang, Victor Weixin, et al. "Mind the gap: Understanding the modality gap in multi-modal contrastive representation learning" Advances in Neural Information Processing Systems 35 (2022): 17612-17625.
>
> > **Q2:** Can the authors analyze how the model's performance on standard benchmarks evolves as the sigmoid contrastive loss progressively approaches zero during training?
>
>
>
> **А:** We set up the following experiment on synthetic data $N = 1000, d=20$ with both trainable relative bias and temperature and optimization using Adam. To track the performance of the optimization we used the margin and the number misclassified pairs (such that $U_i$ is not the closest point to $V_i$ out of the U's or vice verse) of the representations as a proxy (for why this translates to better performance on downstream tasks refer to the answer of Reviewer HL7g's question). This experiment confirmed that the margin continues to improve throughout training and the number of missclassified pairs drops to zero after less than $200$ epochs.
>
>
> > **Q3:** It would be valuable for the authors to offer insights into whether minimizing the sigmoid contrastive loss to zero might adversely affect the model’s generalization ability, e.g. zero-shot retrieval or under query distribution shifts.
>
> **А:**  We don't think this is a big concern for the following two reasons:
> 1.  The loss in SigLIP is already very small. For more on this see the comment under the first question of Reviwer HUkA.
> 2. While over-optimization in deep learning often leads to poor generalization, this is not necessarily the case in our setting. When the sigmoid contrastive loss converges to zero, the resulting representations form an $(m,b_\text{rel})$-constellation—an analytically desirable configuration. Therefore, analyzing the global minima of the loss function is not merely of theoretical interest, but also of practical relevance, as it helps us understand whether the learned representations exhibit the intended structural properties.

---

> > ### Comment · Reviewer_9MsS · 2025-08-04
> >
> > Thanks for the reply. I have read them and will maintain my positive rating.

---

### Official Review · Reviewer_szUN · 2025-07-01

**Clarity:** 2
**Significance:** 1
**Originality:** 2
**Rating:** 4
**Confidence:** 3

**Summary:**

The paper mathematically formalized the synchronization of two representations, mainly discussing when and how the sigmoid-based contrastive objective can drive its loss to zero. Specifically, this paper introduces a new combinatorial structure, the (m,b_{rel}​)-Constellations to show that SigLIP-style losses admit zero-loss solutions precisely when the representations come from one of these constellations. From there they draw out practical implications for retrieval performance, the “modality gap,” embedding dimension requirements, and even propose a cleaner reparameterization of the sigmoid loss.

**Questions:**

Q1: How does “synchronization” differ from “alignment” in latent clustering, and why do synchronization embeddings occupy disjoint regions that will hamper the model performance?


Q2: In the proof of Theorem 3.1, why from 660 to 661, the -\infty becomes the +\infty in the second term when just retrieving t(s)?


Q3: Theorem 3.2 identifies $(m,b_{\rm rel})$-Constellations as all global minima in the t \to\infty, but real training uses finite tt and stochastic optimization. Under finite temperature and sampling noise, do approximate $(m,b_{\rm rel})$-Constellations still govern near-optimal minima, and can we quantify their attraction basins?

Q4: In section 3.4, as the number of modalities k grows or the synchronization graph sparsifiers, how does learning relative biases impact convergence speed and final separation?


Q5: Do you have any performance guarantee for your method in the paper? Like when we set up the hyperparameters which arrive at the upper bound or lower bound as your paper mentioned? Do you have any valid metrics to show the global minimizers improve the embedding?

**Ethical Concerns:**

["NO or VERY MINOR ethics concerns only"]

**Final Justification:**

This work importance is about the "modality gap". I am still trying to understand weather the modality gap will hurt the downstream performance.

**Limitations:**

Yes

**Paper Formatting Concerns:**

No.

**Quality:**

3

**Strengths And Weaknesses:**

For strengths across the dimensions of originality, quality, clarity, and significance:





Quality:




Theoretical insights are rigorously developed, providing a solid foundation for the proposed connections and methodologies.


Clarity:




The manuscript is well structured, with definitions and proofs presented in a logical progression. However, the duplication of “Background” material in both the Introduction and Section 2 could be streamlined to avoid redundancy and sharpen the narrative flow.


Significance:




While the work delivers a compelling geometric explanation for tuning contrastive-loss hyperparameters and formalizes the roots of the “modality gap,” its practical impact hinges on demonstrating how these insights translate into improved performance or design guidelines for real-world multi-modal systems. Clarifying this bridge from theory to real system gains would greatly strengthen the paper’s broader relevance.


Originality:




The paper presents an innovative setup involved the combinatorial characterization of all zero-loss solutions for the sigmoid contrastive objective in the N≫d regime, and showing that these constellations exactly capture when and how one can drive the loss to zero by tuning temperature and bias.



Wakeness:


W1: Although the authors frame their work as a new theory of contrastive losses, the core result—characterizing minimizers of InfoNCE and sigmoid objectives through temperatures and bias—overlaps heavily with existing analyses. It isn’t clear what truly novel insight this paper adds beyond re-deriving known minimizer behavior in a different combinatorial language.


W2:  ImageNet experiments in this paper mainly for visualization of the modality gap through inner‐product histograms, but they are short of measuring end‐to‐end retrieval performance (e.g. top-1/top-5 accuracy) on large-scale benchmarks. It remains unclear whether the theory actually informs practical model design or improvement.



W3: The paper repeatedly invokes the N≫d setting, yet fails to explain why that specific regime generates a modality gap in practice, or why it differs fundamentally from previous high-dimensional results. As a result, the significance of both the regime and the modality gap itself feels under-justified.



W4: While the paper’s ablation study (Section 3.5) systematically compares fixed vs.\ trainable temperature/bias and measures inner‐product margins and loss convergence, some key corollaries—such as “increasing margin m of a constellation makes retrieval robust to larger approximation errors”—are never empirically tested. There are no ablations showing how retrieval performance degrades under quantifiable perturbations or approximate search budgets, leaving theoretical robustness claims unverified.

---

> ### Author Rebuttal · Authors · 2025-07-31
>
> >**W1:** Although the authors frame their work as a new theory of contrastive losses, the core result—characterizing minimizers of InfoNCE and sigmoid objectives through temperatures and bias—overlaps heavily with existing analyses. It isn’t clear what truly novel insight this paper adds beyond re-deriving known minimizer behavior in a different combinatorial language.
>
> **A:**
> We did our best to cite all related work, and are not aware of any theoretical analyses that characterize minimizers in any related setting in the parameter regime of interest. If we have missed any key references, we would be very grateful to the reviewer for a list of papers so that we can appropriately cite them and place our work in better context.
>
> We thoroughly cite prior theoretical work in the field on page 2 (lines 39 - 52). We emphasize that to the best of our knowledge, prior work does not address the practically relevant regime of parameters $d\ll N\ll 2^d,$ and thus, does not explain important qualitative phenomena such as the modality gap (which does not have exist for zero-loss configurations when $N\leq d$). Both our techniques (for example, the use of convex geometry in the proof of Theorem 3.5) and our results are quite different from the ones in prior works. We elaborate when responding to W3 below.
>
>
> >**W2:** ImageNet experiments in this paper mainly for visualization of the modality gap through inner‐product histograms, but they are short of measuring end‐to‐end retrieval performance (e.g. top-1/top-5 accuracy) on large-scale benchmarks. It remains unclear whether the theory actually informs practical model design or improvement.
>
> **A:**
> In Corollary  1 (lines 212 - lines 218), we show that the the inner-product separation condition (i.e., positive margin) of constellations directly implies the success of top-1 retrieval. In this light, our Figure 1 illustrating the inner products in ImageNet is a direct demonstration  of the high success rate of SigLIP on top-1 retrieval. The inner products with matching labels (in blue), which are above all the inner products with mismatched labels (in red), are all correctly classified.
>
> >**W3:**  The paper repeatedly invokes the $N
> \gg d$ setting, yet fails to explain why that specific regime generates a modality gap in practice, or why it differs fundamentally from previous high-dimensional results. As a result, the significance of both the regime and the modality gap itself feels under-justified.
>
>
> **A:** We are not sure which previous high-dimensional results are being referred to; it would be very helpful if the reviewer can provide specific references in order that we can cite and discuss any related work we might have missed.
>
> The practical regime for $N$ and $d$ is $d\ll N \ll 2^d$, since modern machine learning models operate on huge sample sizes ($N$ on the order of 1B-10B) and embed them in a much lower dimension ($d$ is typically between 1k and 10k).
> To the best of our knowledge our work is the first theoretical analysis of sigmoid contrastive loss in the regime $d\ll N \ll 2^d$.
>
> We next describe why the regime $d\ll N$ is closely tied to modality gap.
> Indeed, one of our main theorems (Thm 3.5) demonstrates that it is the SigLIP loss itself that drives configurations to have a modality gap in the parameter regime of interest: zero loss configurations have a modality gap.
> This theorem only holds when $N\ge d + 2$. Indeed, the theorem’s conclusion does not hold otherwise: When $d\ge N+1,$ the configuration $U_i = V_i = X_i$, where $X_1, X_2, \ldots, X_N$ is a simplex, satisfies that $\langle U_i, V_i \rangle > 0 $ for all $i$ and $\langle U_i, V_j \rangle <0$ for all $i\neq j$ (and is therefore a zero-loss configuration), but certainly does not exhibit the modality gap.
>
> On the mathematical side,
> our analysis is substantially different from prior literature, including works demonstrating optimality of simplex configurations (all of which we cite).
> Note that simplices with $N$ points in $d$ dimensions do not even exist when $N>d+1$, and indeed, the minimizing configurations in our setting are very different from a simplex.
>
>
> >**W4:** While the paper’s ablation study (Section 3.5) systematically compares fixed vs.\ trainable temperature/bias and measures inner‐product margins and loss convergence, some key corollaries — such as ``increasing margin m of a constellation makes retrieval robust to larger approximation errors" — are never empirically tested. There are no ablations showing how retrieval performance degrades under quantifiable perturbations or approximate search budgets, leaving theoretical robustness claims unverified.
>
> **A:** Refer to the answer of Reviewer HL7g's question where we explain why larger margin leads to better performance of downstream tasks.
>
> >**Q1:** How does “synchronization” differ from “alignment” in latent clustering, and why do synchronization embeddings occupy disjoint regions that will hamper the model performance?
>
> **A:** We choose to use the term synchronization instead of alignment since perfect alignment between modalities, meaning that $U_i=V_i$ for all $i$, is neither achieved nor necessarily desired. That is, the representations of an image of a cat and the word cat are synchronized, so the corresponding vectors are close, but they are not aligned -- the vectors are still distinct.
>
> The synchronization property holds for virtually all state-of-the-art multimodal models.
> Note that modality gap cannot occur if alignment occurs; in other words, modality gap implies that synchronization rather than alignment occurs.
> Modality gap has been observed empirically by us in SigLIP and modality gap is also observed in CLIP.
>
> We remark that the fact that models are synchronized rather than aligned might actually improve (rather than hamper) performance, since different modalities often contain different information which is best captured by distinct representations.
> An example scenario supporting this intuition was considered in the paper "The Platonic Representation Hypothesis" by Huh et al.
>
> >**Q2:** In the proof of Theorem 3.1, why from 660 to 661, the $-\infty$ becomes the $+\infty$ in the second term when just retrieving t(s)?
>
> **A:** Thank you for catching this typo! It should remain $-\infty.$ This agrees with equation 15 where
> $\langle U_i^{(s)}, V_j^{(s)}\rangle - \text{b}_{\text{rel}}^{(s)}\le 0.$
>
> >**Q3:** Theorem 3.2 identifies $(m, b_\text{rel})$-Constellations as all global minima in the t $\to\infty$, but real training uses finite tt and stochastic optimization. Under finite temperature and sampling noise, do approximate -Constellations still govern near-optimal minima, and can we quantify their attraction basins?
>
> **A:**
> We expect our theoretical insights at infinite temperature to carry over to practically trained models at finite temperature, as we discuss next.
>
> While our main result on the shape of minimizing representation configurations holds in the limit of infinite temperature, we note that practically trained models do have extremely high temperatures: for example, in trained SigLIP models by Google Research, the temperature parameter $t$ is on the order of $100$. As this is inside the exponent defining the loss, a value of 100 is very large, and thus forces behavior on most pairs $\langle U_i, V_j\rangle$ that is well-approximated by infinite temperature.
>
> Empirically, Figs 1 and 2 illustrate that representations of practically trained models share some key geometric aspects with zero loss configurations. We describe these next.
>
> In Fig 1, we show that a trained SigLIP model yields  near perfect separability between inner products of correctly matched pairs and inner products of mismatched pairs. That is,
> $\langle U_i, V_i \rangle \ge \langle U_{i'}, V_{j'}\rangle$ holds for almost all $i, i'\neq j',$ which is exactly the definition of constellations (with $m = (\min_i \langle U_i, V_i\rangle - \max_{i\neq j} \langle U_i, V_j\rangle)/2$ and
> $b_\text{rel} = (\min_i \langle U_i, V_i\rangle + \max_{i\neq j} \langle U_i, V_j\rangle)/2$). This is one crucial aspect in which the geometry of actual representations resembles the geometry of zero-loss configurations.
> In Fig 2, we also showed that trained SigLIP models exhibit the modality gap, a key property of zero-loss configurations.
> > **Q4:** In section 3.4, as the number of modalities $k$ grows or the synchronization graph sparsifiers, how does learning relative biases impact convergence speed and final separation?
>
> We ran additional experiments to investigate how changing the number of modalities affects the margin in the representations (i.e., the final separation) with $N=100$ and $d=10$ and trainable temperature and relative bias. The results are summarized in the table below, more we observe that the margin increases as we increase the number of modalities.
>
> **Final Margins by Number of Modalities**
>
> | Number of Modalities | Final Margin |
> |----------------------|--------------|
> | 2                    | 0.471241     |
> | 4                    | 0.427528     |
> | 6                    | 0.472571     |
> | 8                    | 0.595576     |
> | 14                   | 0.610853     |
> | 20                   | 0.611314     |
>
> >**Q5:** Do you have any performance guarantee for your method in the paper? Like when we set up the hyperparameters which arrive at the upper bound or lower bound as your paper mentioned? Do you have any valid metrics to show the global minimizers improve the embedding?
>
> **A:** A bit of clarification regarding this question would be helpful, we are not sure that we fully understand it.
> As discussed in the paper, we use the margin of a constellation as a proxy for end-to-end retrieval performance. At finite temperature, we have carried out experiments demonstrating that margin increases monotonically as loss decreases. Thus, configurations with lower loss have a larger margin.

---

> > ### Comment · Reviewer_szUN · 2025-08-03
> >
> > I thank the authors for the rebuttal. Now I better understand the significance, as this paper focus on the practically relevant regime $d \ll N \ll 2^d$. I will raise my score to 4, but I still have some questions and would like further clarification regarding the “modality gap”.  Does the modality gap hurt downstream performance?

---

> > > ### Author Response · Authors · 2025-08-03
> > > **Clarification on modality gap vs performance**
> > >
> > > Dear reviewer, Thank you for reading through our rebuttal and for raising your score. We will do our best to clarify how modality gap relates to downstream performance.
> > >
> > > Our paper does not carry out experiments in order to answer the question of whether increasing modality gap improves downstream performance in retrieval tasks, but it does provide some insight as we describe in the next paragraph. We first remark that the paper which introduces the modality gap [1] also states that it is unclear whether the modality gap helps or hinders performance on downstream tasks. Nevertheless, modality gap has been empirically observed in state-of-the-art models ([1] shows this for CLIP, our experiment with ImageNet also confirms this for SigLIP). This suggests that there is no contradiction between modality gap and performance.
> > >
> > > Consider the task of finding an image of a cat given input text 'cat'. For retrieval using top-1 nearest neighbor, the criterion for success is that image of cat is closer to text 'cat' than images of other things. Conversely, If we want to label a cat image, we want that texts associated to cats are closer to image of cat than texts describing other things. Crucially, having a modality gap does not preclude this property, and as mentioned above state-of-the-art models have been empirically observed to have a modality gap. One of the findings of our work is that this is no accident, because there is a rich family of zero-loss configurations of sigmoid contrastive loss with modality gap.
> > >
> > > [1] Liang, Victor Weixin, et al. "Mind the gap: Understanding the modality gap in multi-modal contrastive representation learning" Advances in Neural Information Processing Systems 35 (2022): 17612-17625.

---

> ### Comment · Area_Chair_Txdj · 2025-08-03
> **Regarding your review on novelty**
>
> Dear Reviewer szUN,
>
> Thank you for your detailed review. Could you please elaborate further about your novelty concerns? It would be particularly helpful if you could provide some specific references or examples of prior work. This will give the authors a concrete basis for improving their manuscript and better positioning their contribution.

---

> > ### Comment · Reviewer_szUN · 2025-08-03
> >
> > To be clear, my concern is more about the significance. In lines 63–64 you write that “We show that temperature and bias can drive the loss to zero for a rich family of solutions, which we fully describe in terms of two novel geometric quantities.”
> >
> > As stated, the results about the temperature and bias will lead to the minimizer sounds unsurprising, because prior work (e.g., SimCLR [Chen et al. 2020], Supervised Contrastive Learning [Khosla et al. 2020], and CLIP/SigLIP [Radford et al. 2021]) has already shown that temperature or logit scaling critically shapes the geometry of contrastive minima—affecting alignment vs uniformity and controlling similarity sharpness.
> >
> >
> > 1 Chen, Ting, et al. "A simple framework for contrastive learning of visual representations." International conference on machine learning. PmLR, 2020.
> >
> > 2 Khosla, Prannay, et al. "Supervised contrastive learning." Advances in neural information processing systems 33 (2020): 18661-18673.
> >
> > 3 Radford, Alec, et al. "Learning transferable visual models from natural language supervision." International conference on machine learning. PmLR, 2021.

---

> > > ### Author Response · Authors · 2025-08-03
> > >
> > > > Q: To be clear, my concern is more about the significance. In lines 63–64 you write that “We show that temperature and bias can drive the loss to zero for a rich family of solutions, which we fully describe in terms of two novel geometric quantities.”
> > > As stated, the results about the temperature and bias will lead to the minimizer sounds unsurprising, because prior work (e.g., SimCLR [Chen et al. 2020], Supervised Contrastive Learning [Khosla et al. 2020], and CLIP/SigLIP [Radford et al. 2021]) has already shown that temperature or logit scaling critically shapes the geometry of contrastive minima—affecting alignment vs uniformity and controlling similarity sharpness.
> > > 1 Chen, Ting, et al. "A simple framework for contrastive learning of visual representations." International conference on machine learning. PmLR, 2020.
> > > 2 Khosla, Prannay, et al. "Supervised contrastive learning." Advances in neural information processing systems 33 (2020): 18661-18673.
> > > 3 Radford, Alec, et al. "Learning transferable visual models from natural language supervision." International conference on machine learning. PmLR, 2021.
> > >
> > > A: Thank you for the clarification. We now understand that this sentence can be confusing. We agree with you that it is not a novel observation that trainable temperature parameter helps to reach global minimum (and indeed those references introduce additional trainable parameters to help with optimization).
> > >
> > > The sentence we wrote had two pieces of information to convey. The first is that in both CLIP as well as the original SigLIP paper (and all subsequent papers and implementations of SigLIP) the temperature and bias are trainable parameters. Second, SigLIP has a global minimum of zero (in the limit of large temperature parameter) and this can be an achieved by a rich family of solutions (m,b_r)-Constellations, which we fully describe. (This is one of our results and something which was not known before, especially in the practically relevant setting of $d \ll N \ll 2^d$.)
> > >
> > > We have changed the sentence to: "We show that the SigLIP loss—with trainable temperature and bias—can be driven to zero by a rich family of solutions, which we fully characterize in terms of two novel geometric quantities." We wanted to emphasize that temperature and bias are trainable, because this is a crucial factor that makes the global minimum of SigLIP zero, and it is in this setting that our results hold.

---

### Official Review · Reviewer_HL7g · 2025-07-03

**Clarity:** 3
**Significance:** 3
**Originality:** 3
**Rating:** 5
**Confidence:** 3

**Summary:**

This work theoretically analyzes global minimizers of the Sigmoid loss used to train SigLIP models, and find that resulting embeddings follow a structure termed (m,b_rel)-constellation, which achieves  perfect retrieval and is robust to approximation errors when using approximate nearest neighbor search as the constellation impose a margin between matching and non-matching pairs.

The paper also proves that the modality gap exist in global minimizers and thus show that modality gap does not prevent perfect retrieval.  Based on the theoretical results, the paper proposes a new parametrization of the Sigmoid loss yielding faster training convergence

**Questions:**

Please refer to Strengths And Weaknesses

**Ethical Concerns:**

["NO or VERY MINOR ethics concerns only"]

**Final Justification:**

The paper is technically solid and the rebuttals/discussions answered my questions.

**Limitations:**

Yes

**Paper Formatting Concerns:**

No major formatting issues

**Quality:**

3

**Strengths And Weaknesses:**

The paper is well written and insightful. Previous theoretical works only considered cases where embedding dimension d is higher than number of points N, or number of points N goes to infinity, both cases are impractical, whereas here they consider a more practical setup  d << N << 2^d. Important for practitioners is measuring the evolution of margin during training  as larger margin means more robustness to errors in approximate nearest neighbor search.

The relative bias parametrization is only shown to be faster convergence than original Sigmoid loss on a toy example with N=100 and d=10 and a single run with fixed hyper-parameters, which makes the conclusion limited. How does it behave on a real world example ? and does the faster convergence also translate to better performance on downstream tasks (e.g., zero-shot classification) ?

---

> ### Author Rebuttal · Authors · 2025-07-31
>
> > **Q:** The relative bias parametrization is only shown to be faster convergence than original Sigmoid loss on a toy example with N=100 and d=10 and a single run with fixed hyper-parameters, which makes the conclusion limited. How does it behave on a real world example ? and does the faster convergence also translate to better performance on downstream tasks (e.g., zero-shot classification) ?
>
> **A:** To address the question about scaling, we ran a larger experiment with $N = 50,000$ synthetic examples ($10^5$ data points total) and dimension $d = 100$ the results are qualitatively the same as the ones described in the paper which is good evidence that the empirical results are scalable.
>
> Beyond our empirical results concerning relative bias, we emphasize that our main contributions are to carry out theoretical analysis characterizing the minimizers of sigmoid contrastive loss in the practically relevant parameter regime. Prior to our work it was not at all clear what sorts of configurations this loss seeks. We further determined a number of interesting properties both of theoretical and practical interest of the minimizing configurations, including modality gap and perfect retrieval via 1-top nearest neighbor search.
>
> While we have not performed experiments confirming this hypothesis, based on prior work on representation synchronization (albeit with different contrastive loss functions), it is to be expected that larger margins, which are achieved by the relative bias parameterization, lead to better performance in various downstream tasks. For example:
>
> 1.  "Understanding Contrastive Representation Learning through
> Alignment and Uniformity on the Hypersphere" by Tongzhou Wang and Phillip Isola (e.g. Tables 1 an 2) show how directly optimizing the representation properties of alignment (which corresponds to a smaller modality gap) and uniformity (which corresponds to higher margin because higher margin means more uniform packing of the sphere with caps of that margin) lead to better downstream performance on classification (CIFAR-10 and STL-10) and indoor scene understanding (NYU-Depth-V2).
>
> 2. "ArcFace: Additive Angular Margin Loss for Deep Face Recognition" which shows that directly optimizing the margin in the representations leads to better downstream performance on face recognition tasks.
>
> 3. "CoReS: Compatible Representations via Stationarity" which was pointed out to us by Reviewer HUkA also uses the heuristic that larger margin means better performance by embedding the classes to a regular polytope. While their work requires the dimension of the representation space to be very high (one lower than the number of classes) it still shows us that larger margins lead to better performance.

---

### Official Review · Reviewer_HUkA · 2025-07-03

**Clarity:** 3
**Significance:** 4
**Originality:** 4
**Rating:** 4
**Confidence:** 4

**Summary:**

This paper provides a rigorous theoretical analysis of the sigmoid contrastive loss, a key component in modern large-scale vision-language models like Google's SigLIP. The authors identify a gap between the practical success of these models and the theoretical understanding of their underlying mechanisms, particularly in the realistic regime where the number of training examples (N) far exceeds the embedding dimension (d) but is not infinite.
The paper characterize the global minima of the sigmoid using (m, b_rel)-costellations, geometrical comfigurations which correspon to the spherical codes on a hypersphere.
Specifically, the Relative Bias Parameterization is proposed to address the problem of the optimization dynamics which naturally force the effective relative bias (b/t) to zero as t grows large.
Using this framework, the paper offers theoretical explanations for the modality gap phenomenon and sunchronization of multiple modalities.

**Questions:**

1. To facilitate the adoption of L_RB-Sig, could you provide practical guidance on initializing the learnable parameters? Specifically, what initial values for t and b_rel would you recommend for stable large-scale training?
2. For the case of a fixed b_rel, how sensitive is the model's performance to this choice? What is the impact of choosing a suboptimal value compared to both the learned b_rel and the standard L_Sig baseline?
3. Your theory is based on ideal zero-loss configurations. How do you expect the geometric properties of the (m, b_rel)-constellations to behave in a realistic, non-zero loss regime typical of large-scale training?

**Ethical Concerns:**

["NO or VERY MINOR ethics concerns only"]

**Final Justification:**

The authors have clarified all my points and considering also the other reviews I suggest the acceptance for this paper.

**Limitations:**

Yes, in Sec 4.

**Paper Formatting Concerns:**

- Typo in L349 ``teh''
- Be consistent when you refer to the Construction 1 you refer to it as Example 1 (eg in L243)

**Quality:**

4

**Strengths And Weaknesses:**

**Strengths**:
- The paper's greatest strength is its formal approach. By introducing concepts from geometry and coding theory like (m, b_rel)-constellations and spherical codes, it brings mathematical precision to a field often driven by empirical results. This provides a solid foundation for understanding why these models work.
- The work is not just abstract theory. It is directly motivated by and analyzes state-of-the-art models (SigLIP). It successfully explains real-world phenomena like the modality gap, which had previously been an empirical observation without a strong theoretical justification.
- The proposed L_RB-Sig is a simple, practical, and elegant solution to the identified problem. It's a minor, well-justified change to the loss function that has significant theoretical and practical benefits. The paper also provides clear guidance on how to use it (either learning b_rel or freezing it to guide the learned geometry of the space).



**Weaknesses**:

While theoretically strong, the paper's practical impact is limited by several key weaknesses. It often priorites the ``what'' over the ``why'' letting to the reader the effort to concretically understand the meaning of the theory:

- The paper overlooks the related field of backward-compatible representation learning (eg [1,2,3,4]) is missing in the synchronization of representations even though the aim is the same
- The theoretical analysis primarily characterizes the geometry of "global minimizers," which are configurations that achieve zero loss. In practice, training on massive, noisy web-scale data never results in zero loss. While the principles likely still hold, the analysis doesn't fully address how these geometric structures are affected in the more realistic, non-zero loss regime. Are the constellations easy to constryct also in this case?
- The paper concludes that the larger margins achieved by L_RB-Sig mean the representations are "more robust on retrieval tasks." This is a well-accepted and highly plausible heuristic in the field, but without a direct evaluation of a downstream retrieval task, this remains an assumed (though likely correct) consequence rather than a demonstrated fact of the paper.
- Adapters are proposed and never evaluated. Furthermore, a comparison with a baseline with a standard trainable adapter (e.g. [3,4,5]) should be included.
- The paper lacks crucial implementation details that would help practitioners adopt its findings. For example, it provides no pseudocode or guidance on how to initialize the learnable t and b_rel parameters, which is critical for stable training.

[1] Shen, Y., Xiong, Y., Xia, W., & Soatto, S. (2020). Towards backward-compatible representation learning. In Proceedings of the IEEE/CVF Conference on Computer Vision and Pattern Recognition (pp. 6368-6377).

[2] Biondi, N., Pernici, F., Bruni, M., & Del Bimbo, A. (2023). Cores: Compatible representations via stationarity. IEEE Transactions on Pattern Analysis and Machine Intelligence, 45(8), 9567-9582.

[3] Ramanujan, V., Vasu, P. K. A., Farhadi, A., Tuzel, O., & Pouransari, H. (2022). Forward compatible training for large-scale embedding retrieval systems. In Proceedings of the IEEE/CVF Conference on Computer Vision and Pattern Recognition (pp. 19386-19395).

[4] Jaeckle, F., Faghri, F., Farhadi, A., Tuzel, O., & Pouransari, H. (2023). Fastfill: Efficient compatible model update. ICLR2023.

[5] Moschella, L., Maiorca, V., Fumero, M., Norelli, A., Locatello, F., & Rodolà, E. (2022). Relative representations enable zero-shot latent space communication. ICLR2023.

---

> ### Author Rebuttal · Authors · 2025-07-31
>
> >**Q1:** To facilitate the adoption of $\mathcal{L}^{RB-Sig}$, could you provide practical guidance on initializing the learnable parameters? Specifically, what initial values for t and $b_\text{rel}$ would you recommend for stable large-scale training?
>
> **A:** Our theoretical analysis currently does not yield any initialization recommendations. In experiments with synthetic data (where we optimize directly over the representations rather than a given network), it seems like the initial choice of $b_\text{rel}$ makes little difference when it is trainable.
>
> We point out that in the original SigLIP paper ``Sigmoid Loss for Language Image Pre-Training'' by Xiaohua Zhai, Basil Mustafa, Alexander Kolesnikov, Lucas Beyer, there is a recommendation for $t = 10, b = -10$ (corresponding to $b_\text{rel} = -1$). When using these initial values for our experiment with trainable relative bias we always converge to $0$ loss and stable training. The setting we used is $N=1024$ and $d=20$ using Adam with both $t$ and $b_\text{rel}$ trainable with the representations initialized i.i.d. uniformly on the sphere. We ran a hyperparameter search to find the best initialization for $t \in \{ 1,3,10,30,100\}$ and $b_\text{rel} \in \{ -1.0, -0.8, \dots, 0.8, 1.0\}$ and found that in general the best representations (measured by the final margin $\min_{i} (U_i, V_i) - \max_{i\neq j}(U_i, V_j)$) is achieved for small $t  \lesssim 3$ or intermediate $t\approx 10$ and  $b_\text{rel} \gtrsim 0.6$. In those settings the final relative bias was $\approx 0.7$.
>
> A possible intuition justifying starting with small $t$ is that the representations can move relatively freely when $t$ is small, and only once they arrive at a configuration attaining small loss does the temperature parameter start to increase, further driving down the loss.
>
> The margins of final configuration are in the following table, with rows indexed by initial t and columns by initial $b_\text{rel}$:
> \\begin{array}{c|rrrrrrrrrrr}
> \\text{Temp} & -1.0 & -0.8 & -0.6 & -0.4 & -0.2 & 0.0 & 0.2 & 0.4 & 0.6 & 0.8 & 1.0 \\\\
> \\hline
> 1   & 0.567 & 0.567 & 0.566 & 0.564 & 0.566 & 0.566 & 0.565 & 0.568 & 0.570 & \\mathbf{0.574} & 0.573 \\\\
> 3   & 0.545 & 0.543 & 0.526 & 0.499 & 0.483 & 0.488 & 0.536 & 0.563 & 0.570 & 0.573 & \\mathbf{0.574} \\\\
> 10  & 0.439 & 0.425 & 0.415 & 0.406 & 0.402 & 0.410 & 0.429 & 0.524 & 0.566 & 0.547 & 0.562 \\\\
> 30  & 0.301 & 0.297 & 0.294 & 0.315 & 0.343 & -0.942 & -0.915 & -0.935 & -1.171 & -1.051 & -1.145 \\\\
> 100 & -0.774 & -0.679 & -0.483 & -0.740 & -0.878 & -0.956 & -0.978 & -1.109 & -1.186 & -1.080 & -1.449 \\\\
> \\end{array}
>
> >**Q2:** For the case of a fixed $b_\text{rel}$, how sensitive is the model's performance to this choice? What is the impact of choosing a suboptimal value compared to both the learned $b_\text{rel}$ and the standard $L^{Sig}$ baseline?
>
> **A:**
> In our experiments given below in Table 2, we investigated the impact of the fixed value of $b_\text{rel}$. We initialized the $N=1024$ points i.i.d. uniformly at random on the sphere in $d=20$ dimensions, and for the initial temperature we chose $t = 10$. Runs were stopped whenever gradients of all representations dropped below $1e-10$ or after $10k$ epochs.  We see that a range of $b_\text{rel}$ between $-0.2$ and $0.7$ yield nearly zero loss when training stabilizes. The size of the margin does, however, seem to depend fairly significantly on the value of $b_\text{rel}$, as seen in the table.
>
> Choosing $b_\text{rel}\approx 0.7$ yields the largest final margin, and this matches the final value for trainable $b_\text{rel}$ in the experiments described above in response to Q1. Other choices for fixed $b_\text{rel}$ result in smaller final margins (and in the extremes, larger final loss) as indicated in the following table.
>
> **Table 2**: Rows: Fixed Relative Bias
>
> | Fixed Relative Bias | Final Temperature | Achieved Margin | Final Loss |
> |---------------------|-------------------|------------------|-------------|
> | -1.000000           | 6.961601          | -0.000001        | 0.693150    |
> | -0.900000           | 56.188858         | -0.000000        | 0.009245    |
> | -0.800000           | 23.300198         | -0.000000        | 0.014105    |
> | -0.700000           | 162.664841        | 0.092437         | 0.000005    |
> | -0.600000           | 125.656731        | 0.122326         | 0.000003    |
> | -0.500000           | 104.095329        | 0.152893         | 0.000003    |
> | -0.400000           | 90.788620         | 0.182992         | 0.000002    |
> | -0.300000           | 81.079422         | 0.213618         | 0.000001    |
> | -0.200000           | 75.061546         | 0.242438         | 0.000001    |
> | -0.100000           | 71.254242         | 0.273600         | 0.000000    |
> | 0.000000            | 69.018265         | 0.301340         | 0.000000    |
> | 0.100000            | 69.534515         | 0.329022         | 0.000000    |
> | 0.200000            | 67.996437         | 0.353406         | 0.000000    |
> | 0.300000            | 61.796310         | 0.390087         | 0.000000    |
> | 0.400000            | 55.452553         | 0.430921         | 0.000000    |
> | 0.500000            | 48.406261         | 0.466707         | 0.000000    |
> | 0.600000            | 44.391388         | 0.498564         | 0.000000    |
> | 0.700000            | 42.265457         | 0.527834         | 0.000000    |
> | 0.800000            | 37.361767         | 0.539749         | 0.000001    |
> | 0.900000            | 33.167175         | 0.483351         | 0.000036    |
> | 1.000000            | 23.817665         | 0.513416         | 0.000693    |
>
> >**Q3:** Your theory is based on ideal zero-loss configurations. How do you expect the geometric properties of the $(m, b_\text{rel})$-constellations to behave in a realistic, non-zero loss regime typical of large-scale training?
>
> **A:** We find that representations of practically trained models  (at non-infinite temperature) share multiple key geometric aspects with zero loss configurations at infinite temperature parameter. We elaborate on this in a moment.
>
> We first remark that while our main result is for zero-loss configurations in the limit of infinite temperature, our experiments (e.g. in Table 2 above) do result in nearly zero loss. Additionally, practically trained models end up with very high temperatures: e.g., in trained SigLIP models by Google Research, the final temperature parameter $t$ is on the order of $100$.
> Because this is in the exponent of the loss, the resulting configurations are well-approximated by infinite temperature zero-loss configurations: pairs $\langle U_i, V_j\rangle$ leading to non-zero loss at infinite temperature will have a large contribution also for $t=100$.
>
>
> We next discuss how Figures 1 and 3 illustrate that representations of practically trained models share key geometric aspects with zero loss configurations.
> Figure 1 shows that a trained SigLIP model yields  near perfect separability between inner products of correctly matched pairs and inner products of mismatched pairs. That is,
> $\langle U_i, V_i \rangle \ge \langle U_{i'}, V_{j'}\rangle$ holds for almost all $i, i'\neq j',$ which is exactly the definition of constellations (with $m = (\min_i \langle U_i, V_i\rangle - \max_{i\neq j} \langle U_i, V_j\rangle)/2$ and
> $b_\text{rel} = (\min_i \langle U_i, V_i\rangle + \max_{i\neq j} \langle U_i, V_j\rangle)/2$). This is one crucial aspect in which the geometry of actual representations resembles the geometry of zero-loss configurations.
>
> Figure 3 shows that trained SigLIP models exhibit the modality gap, a property shared by zero-loss configurations as described in Theorem 3.5.
>
> Thus, these figures and experiments demonstrate that our theoretical insights indeed carry over to practically trained models.
>
>
> >**W1 and W3 (paraphrased):** The paper overlooks previous work on backward-compatible representation learning.
>
> **A:** Thank you for pointing out the line of work on  backward-compatible representation learning. This direction is highly relevant to our paper and aligns with the broader set of applications we mention, such as modality synchronization and distillation. We have added the appropriate references to acknowledge this connection.
>
> Briefly, our work fits into that literature as follows. Our primary focus is not on benchmarking specific applications of contrastive learning. Instead, our work aims for a theoretical understanding of phenomena appearing from training with the sigmoid contrastive loss. As far as we are aware, such theoretical analysis does not exist for any contrastive learning objective in the $N\gg d$ regime of interest. We hope to offer practitioners tools, such as the relative bias parameter, to modulate their behavior. We view backward-compatible representation learning as a particularly promising application of our findings and an exciting avenue for future research.
>
>
> >**W2:** The theoretical analysis primarily characterizes the geometry of ``global minimizers," which are configurations that achieve zero loss. In practice, training on massive, noisy web-scale data never results in zero loss. While the principles likely still hold, the analysis doesn't fully address how these geometric structures are affected in the more realistic, non-zero loss regime. Are the constellations easy to construct also in this case?
>
> **A:** Please refer to the answer of Q3.

---

### Decision · Program_Chairs · 2025-09-17

**Decision:**

Accept (poster)

**Comment:**

This paper presents a theoretical analysis of the sigmoid contrastive loss, particularly in the practical regime where the number of data points (N) is much larger than the embedding dimension (d). The authors introduce (m, rb)-Constellations, a geometric structure, to characterize the set of global minimizers for this loss function when temperature and bias are trainable. The paper's primary strength, as noted by all reviewers, is its rigorous theoretical approach to understanding a widely used loss function. It provides a solid mathematical foundation for empirical phenomena observed in state-of-the-art models like SigLIP, most notably offering a formal explanation for the "modality gap." The work is well-motivated and directly connects its theoretical findings to practical aspects of representation learning, such as retrieval robustness and multi-modality synchronization.

The main weaknesses, raised consistently by reviewers HUkA, HL7g, and szUN, center on the limited empirical validation of the paper's practical claims. For instance, the proposed re-parameterization of the sigmoid loss (L_RB-Sig) is shown to have faster convergence, but only on small-scale synthetic data. Claims about improved retrieval robustness due to larger margins are theoretically justified but not demonstrated on downstream tasks. During the rebuttal period, the authors addressed these points by arguing that their zero-loss analysis is a reasonable approximation for real-world models which train to very high temperatures. They also clarified the novelty of their work by highlighting its focus on the d << N regime, which distinguishes it from prior analyses and led reviewer szUN to raise their score. The authors acknowledged the limited scope of their experiments but provided additional synthetic results and cited relevant literature to bridge the gap between their theoretical contributions and expected practical outcomes.

Despite the borderline scores, the paper's theoretical contributions are significant enough to warrant acceptance. The characterization of global minimizers via (m, rb)-Constellations and the principled explanation of the modality gap are novel insights that will be of considerable interest to the representation learning community. The discussion period clarified the scope and significance of the work. Therefore, I recommend this paper for acceptance. The authors are strongly encouraged to incorporate the feedback from the reviewers, particularly by better contextualizing the practical implications of their theoretical results and acknowledging the limitations of their experimental validation in the final version.